# Grounded Reinforcement Learning
# for Visual Reasoning

**Gabriel Sarch**     **Snigdha Saha**     **Naitik Khandelwal**     **Ayush Jain**

**Michael J. Tarr**     **Aviral Kumar**     **Katerina Fragkiadaki**

Carnegie Mellon University

visually-grounded-rl.github.io

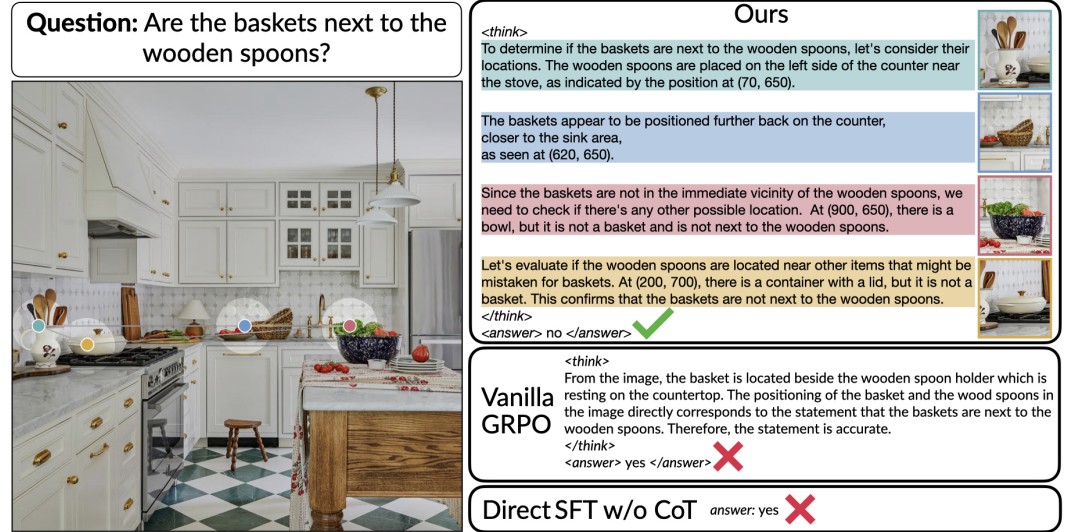

Figure 1: **Grounded visual reasoning enables interpretable and accurate answers.** ViGoRL decomposes the task into a sequence of natural language thoughts anchored in image regions. In contrast, Vanilla GRPO and SFT baselines produce ungrounded and incorrect responses.

## Abstract

While reinforcement learning (RL) over chains of thought has significantly advanced language models in tasks such as mathematics and coding, visual reasoning introduces added complexity by requiring models to direct visual attention, interpret perceptual inputs, and ground abstract reasoning in spatial evidence. We introduce ViGoRL (**Vi**sually **Gro**unded **R**einforcement **L**earning), a vision-language model trained with RL to explicitly anchor each reasoning step to specific visual coordinates. Inspired by human visual decision-making, ViGoRL learns to produce spatially grounded reasoning traces, guiding visual attention to task-relevant regions at each step. When fine-grained exploration is required, our novel multi-turn RL framework enables the model to dynamically zoom into predicted coordinates as reasoning unfolds. Across a diverse set of visual reasoning benchmarks—including SAT-2 and BLINK for spatial reasoning, V*bench for visual search, and ScreenSpot and VisualWebArena for web-based grounding—ViGoRL consistently outperforms both supervised fine-tuning and conventional RL baselines that lack explicit grounding mechanisms. Incorporating multi-turn RL with zoomed-in visual feedback significantly improves ViGoRL's performance on localizing small GUI elements and visual search, achieving 86.4% on V*Bench. Additionally, we find that grounding amplifies other visual behaviors such as region exploration, grounded subgoal setting, and visual verification. Finally, human evaluations show that the model's visual references are not only spatially accurate but also helpful for understanding model reasoning steps. Our results show that visually grounded RL is a strong paradigm for imbuing models with general-purpose visual reasoning.

39th Conference on Neural Information Processing Systems (NeurIPS 2025).

# 1 Introduction

Visual reasoning tasks vary widely in structure and often demand different solution strategies depending on the problem at hand. Some tasks are dominated by salient visual cues, such as recognizing a refrigerator centered in a kitchen scene, while others, like locating a pair of scissors in a cluttered environment, require sequential visual search and selective attention. Despite this diversity, most state-of-the-art vision-language models (VLMs) operate in an end-to-end fashion, predicting answers directly in a single forward pass. These models often lack the ability to adapt their computational strategies to different tasks or to expose intermediate reasoning beyond visual attention maps. Prompt-based models, such as ViperGPT [54], VisualProg [22], and V* [66], explicitly decompose visual tasks into sequences of subgoals or intermediate steps to improve interpretability and performance without additional training. However, such models typically generate fixed reasoning chains that do not adapt to the structure of the input scene.

Recent advances in reinforcement learning (RL) over reasoning chains have significantly enhanced the capabilities of LLMs in text-based domains [21, 57, 28], enabling them to learn diverse reasoning strategies tailored to the context. However, RL can only build upon skills or compose reasoning behaviors that are already latent in the base model's sampling distribution [17, 81]. For e.g., Gandhi et al. [17] has identified key cognitive behaviors in text-based domains, such as setting sub-goals, backtracking, verification, that support self-improvement under RL. Models lacking these behaviors often do not benefit from RL and must be bootstrapped via supervised fine-tuning (SFT) on curated reasoning traces before RL is run [17, 81]. However, it remains unclear whether the cognitive behaviors identified in text-based domains similarly support generalization in visual reasoning tasks.

Several recent works have attempted RL fine-tuning directly on base vision-language models (VLMs) [40, 39, 86, 55, 57, 83, 51, 4, 34], implicitly assuming RL alone can induce useful cognitive behaviors. However, our analysis reveals that such naïve applications of RL typically yield abstract, ungrounded reasoning rather than richer, visually grounded cognitive behaviors (see Section 3.1, 5.3). These findings align with prior research showing that explicitly prompting VLMs to reference spatial object locations improves performance and interpretability [66, 72], suggesting that grounding thought in spatial regions may serve as a key cognitive behavior for effective visual reasoning. Thus, a critical open question arises: *How can we embed useful cognitive behaviors in VLMs before applying RL to achieve robust visual reasoning?*

We hypothesize that models both "see better" and "think better" when their textual reasoning steps are explicitly grounded in specific image regions, promoting more targeted and systematic cross-referencing between textual and visual information during reasoning. This hypothesis is inspired by the fact that humans systematically shift their restricted gaze to selectively gather and integrate task-relevant information when reasoning about the world [74, 25, 73]. Grounding may serve a similar role in models, functioning as a spatial attention mechanism that enables accurate feature binding [60, 7, 8] and supports deictic reference [6] to simplify multi-step reasoning through localized perceptual anchoring. We move beyond prompt-based reasoning, proposing that learning to compose *reasoning steps explicitly anchored in image coordinates* induces structured region-level behaviors that support improved generalization in visual tasks.

**Our Approach.** We introduce a multi-turn RL framework for training VLMs to reason in a grounded, visually-aware manner. This stands in contrast to LLM reasoning in math or code, where grounding in external input is not strictly required. Within each reasoning step, the model produces a natural language thought along with a corresponding spatial grounding (i.e., an $(x, y)$ location in the image). This enables it to progressively refine its attention and gather task-relevant visual information as reasoning unfolds. By incorporating multi-turn interaction into the RL process—where each turn consists of one or more reasoning steps followed by a query to a visual feedback tool—the model learns to iteratively request zoomed-in views of selected regions when fine-grained visual information is required. Critically, no external supervision or explicit human-provided grounding cues are used to supervise the spatial grounding of the thought; instead, the model autonomously learns to propose and utilize spatial grounding as an internal cognitive tool.

Current methods for training VLMs to directly produce textual answers from visual inputs inherently bias them toward abstract, ungrounded reasoning, making it fundamentally difficult for RL methods alone to spontaneously discover systematic visual strategies at the region-level. To explicitly inject grounded reasoning behaviors before RL training, we employ Monte Carlo Tree Search (MCTS) to systematically stitch together independently sampled reasoning steps, generating diverse, visually-

grounded reasoning trajectories. We bootstrap the model via supervised fine-tuning (SFT) on these MCTS-constructed paths, thus embedding rich region-level reasoning strategies into the model. We then apply RL, through Group Relative Policy Optimization (GRPO) [50], to further reinforce grounded sequences that lead to correct answers. Finally, we introduce a novel multi-turn RL formulation with visual feedback loops, allowing the model to dynamically zoom into image regions via tool calling for more detailed visual inspection when needed. This multi-turn variant of our method improves the model's capacity to localize and reason about fine-grained visual elements.

**Empirical Results.** We evaluate ViGoRL across a suite of visual reasoning benchmarks, including SAT-2 [48], BLINK [15], RoboSpatial [53], ScreenSpot [10, 33], VisualWebArena [31], and V*Bench [66]. Our approach consistently outperforms existing methods on all tasks. Specifically, ViGoRL achieves substantial improvements over vanilla GRPO, with accuracy gains of 12.9 points on SAT-2 and 2.0 points on BLINK. In fine-grained web grounding scenarios, our method surpasses both vanilla GRPO and large-scale web-finetuned models on ScreenSpot-Pro. By leveraging multi-turn RL for dynamic, zoomed-in visual feedback, ViGoRL further improves performance on ScreenSpot-Pro, effectively localizing small elements within high-resolution images. Moreover, multi-turn RL significantly enhances visual search capabilities, allowing ViGoRL to outperform both VLM tool-use pipelines and proprietary VLMs on V*Bench, achieving an accuracy of 86.4%. On VisualWebArena, a benchmark requiring live web interaction from image inputs alone, without access to HTML, ViGoRL outperforms both direct SFT and vanilla GRPO, and surpasses the previous state-of-the-art for this model size, ICAL [49], despite using only visual input.

Ablation studies confirm the importance of grounding: models trained without spatial anchoring perform significantly worse. Further, we find that grounding amplifies other visual cognitive behaviors such as region exploration, goal setting, and visual verification. Human evaluations show that our model's reasoning is both spatially accurate and helpful to understanding the model's reasoning steps. Our results point to visually grounded RL as a strong paradigm for general-purpose visual reasoning.

## 2 Related Work

**Programmatic Reasoning in VLMs.** Vision-Language Models (VLMs) [46, 29, 3, 9, 62, 56] excel on multimodal tasks through large-scale pretraining, but struggle with complex reasoning such as counting [47], spatial reasoning [48], and compositional understanding [59]. Prompting strategies like chain-of-thought (CoT) [64, 32], mm-CoT [85], IoT prompting [90], and Mind's Eye [68] guide models to generate explicit reasoning steps grounded in images. Methods such as V* [66], Sketchpad [26], VisProg [22], REFOCUS [16], and ViperGPT [54] use language models to produce executable visual plans but rely on frozen backbones and hand-crafted prompts.

**Distillation and Supervised Fine-Tuning.** To robustly instill reasoning skills, supervised fine-tuning (SFT) methods train on curated reasoning trajectories. For text tasks, STaR [82] generates reasoning via few-shot prompting and selects based on correctness, while S1 [41] distills reasoning chains into smaller models. Similar approaches in VLMs include LLAVA-CoT [71], which distills CoT reasoning from GPT-4o [1], and ICAL [49], VPD [27], and Mulberry [76], which employ LLMs or MCTS to generate reasoning data. Methods like VOCOT [35] improve grounding of entities via SFT. However, distillation methods rely solely on positive examples, neglecting failed paths. RL addresses this by learning from both successes and failures, outperforming SFT alone in our experiments.

**Reinforcement Learning on Chains of Thought.** Applying RL to chains of thought has improved reasoning in verifiable domains like math and coding [28, 21, 57]. Early methods [84, 43, 14, 67, 20] iteratively refine reasoning using preference-based approaches, while recent efforts like DeepSeek-R1 [21] and Kimi-1.5 [57] leverage online RL with outcome-based rewards. While initially believed to induce novel cognitive behaviors, analyses [17, 81, 38, 78] suggest RL primarily amplifies existing capabilities already found in the base model. Most prior work focuses on text-only domains, leaving visual reasoning behaviors largely unexplored. Visual-RFT [39] applies RL to textual reasoning in VLMs without incentivizing new visual behaviors. In contrast, we explicitly ground reasoning steps visually, amplifying exploration, verification, and backtracking via learned visual interactions.

# 3 Preliminaries

**Reasoning in Vision-Language Models.** The overarching objective of our work is to improve the reasoning capabilities of vision-language models (VLMs). We consider reasoning tasks defined by a dataset $\mathcal{D}$ of problem instances $(I, q, a^*)$, where $I$ is a visual input (e.g., an image), $q$ is a natural language query about the image, and $a^*$ is the correct, verifiable answer (e.g., a class label, bounding box, or discrete actions such as `click [element]`). The goal is to train a vision-language policy $\pi_\theta$ parameterized by $\theta$ that outputs a reasoning trace $\tau$ consisting of sequential textual reasoning steps $s_1, s_2, \ldots, s_T$ culminating in an answer $a$. This policy factorizes autoregressively:

$$\pi_\theta(\tau \mid I, q) = \left(\prod_{t=1}^{T} \pi_\theta(s_t \mid I, q, s_{<t})\right) \cdot \pi_\theta(a \mid I, q, s_{\leq T}).$$

While supervised fine-tuning can teach models to mimic reasoning chains provided in training data, RL offers the potential to directly reinforce reasoning behavior sampled from the base model based on correctness or other reward signals [42, 50, 17]. Specifically, RL allows us to optimize policies over reasoning traces $\tau$ that maximize expected returns based on task performance and adherence to desired format structure. Formally, the RL objective can be expressed as:

$$\max_\theta \; \mathbb{E}_{(I,q,a^*)\sim\mathcal{D}} \left[\mathbb{E}_{\tau\sim\pi_\theta}[R(\tau)]\right], \quad \text{s.t. } \tau \text{ satisfies format constraints,}$$

where the reward $R(\tau)$ typically includes correctness of the final answer, and proper adherence to structured reasoning formats.

## 3.1 Do Current RL Recipes Amplify VLM Behaviors That Support Visual Reasoning?

It has been shown that RL on chains of thought alone cannot necessarily induce new behaviors from scratch; it can only amplify or chain reasoning primitives that are already present in the base model's sampling distribution [17, 81]. Do current base VLMs exhibit desirable visual reasoning behaviors and can RL amplify these behaviors to improve performance? Following previous work from LLMs [17], we categorize the behaviors of base VLMs when tasked with the Spatial Aptitude Test [48] spatial reasoning benchmark, which requires the synthesis of evidence in multiple regions of one or multiple images to answer. Our complete analysis and experimental protocol is detailed in Section 5.3 and Appendix A6. Our analysis reveals two key insights:

- *Takeaway 1: Current VLMs often fail to reference fine-grained image inputs; their reasoning is largely ungrounded.* Without explicit grounding, models treat vision as static context rather than actively referenced input, spending tokens on abstract thought instead of analyzing visual regions. Qwen2.5-VL-3B examines only 1.44 regions per task with minimal visual verification (0.14) and no backtracking. We provide an examples of such an output in Figure 2.

- *Takeaway 2: Standard RL optimization exacerbates ungrounded reasoning.* RL-tuning with task-level rewards slightly increases region exploration (1.8) but eliminates visual subgoal setting (0.00) and still shows no backtracking (0.00). Figure 2 and Appendix A9.1 illustrates typical outputs under this setup. This demonstrates that optimizing for correctness without encouraging grounded reasoning fails to instill visual reasoning skills. Furthermore, prompt engineering for broader visual behaviors is ignored during RL training (Section 5.2, Table 4).

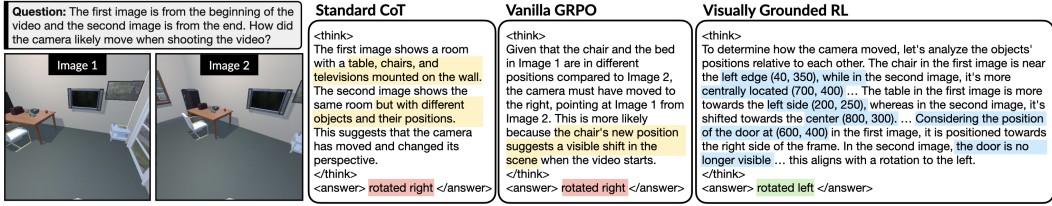

Figure 2: Without actively reinforcing visually grounded behaviors, RL collapses onto shortcuts that maximize immediate rewards at the expense of richer visual reasoning. **Standard CoT** and **Vanilla GRPO** (left and center) exhibit visually ungrounded reasoning, relying on vague references to scene elements (shown in yellow), which often results in incorrect answers (marked in red). In contrast, **Visually Grounded RL** (right) explicitly references object positions, demonstrating precise spatial grounding (shown in blue) and more often producing correct reasoning outcomes (marked in green). See Section 5.3 for further analysis.

**Why does standard RL fail here?** We hypothesize that the failure mode of RL comes down two interrelated issues: (1) the initial sampling distribution of pretrained VLMs heavily biases the model toward abstract, language-based strategies rather than region-level analysis (as shown in Takeaway 1 above), (2) since standard RL provides rewards based solely on final correctness and general formatting, it amplifies behaviors that attain high rewards irrespective of how they actually maximize reward. Without actively biasing the initial policy and reinforcing visually grounded behaviors, RL naturally collapses onto shortcuts that maximize immediate rewards at the expense of richer visual reasoning. *Thus, we hypothesize that encouraging explicit grounding biases the model's reasoning distributions toward exploring relevant visual regions, setting meaningful visual subgoals, verifying visual hypotheses, and effectively backtracking.*

## 4 Visually Grounded Reinforcement Learning (ViGoRL)

The analysis in Section 3.1 shows that naïve RL on small VLMs may degrade into ungrounded reasoning strategies, and that incentivizing grounding may improve visual reasoning behaviors that improve generalization. To incorporate explicit visual grounding into the reasoning process, we redefine each reasoning step as a tuple: $n_t = \langle s_t, (x_t, y_t) \rangle$, where $s_t$ is a textual thought and $(x_t, y_t)$ anchors it to a specific image location. The full trajectory becomes $\tau = [n_1, \ldots, n_T, a]$. This modifies the original factorization from Section 3:

$$\pi_\theta(\tau \mid I, q) = \left( \prod_{t=1}^{T} \pi_\theta(n_t \mid I, q, n_{<t}) \right) \cdot \pi_\theta(a \mid I, q, n_{\leq T}),$$

where each $n_t$ now includes both the reasoning step and its visual grounding. By introducing this grounding constraint, we explicitly guide the model to systematically reference specific image locations as evidence for its reasoning. This grounding incentivizes the model to iteratively explore and verify distinct visual regions, formulate and ground intermediate subgoals visually, and revisit prior regions when uncertainty or errors arise.

### 4.1 Our Approach for Grounded Reinforcement Learning

Building on our findings in Section 3.1, we introduce a comprehensive approach that directly addresses the ungrounded reasoning patterns in VLMs. We propose a two-stage pipeline to incorporate grounded reasoning as detailed in the previous section: **(1) warm-start supervised finetuning** that biases the model toward generating structured reasoning chains with explicit spatial grounding, followed by **(2) reinforcement learning** that systematically refines these grounded behaviors. Finally, we extend our approach to multi-turn RL (Section 4.3.1), enabling fine-grained visual feedback at each reasoning step. This pipeline yields models that invest test-time compute into examining diverse image regions—precisely the behaviors our analysis showed were absent in standard VLM reasoning.

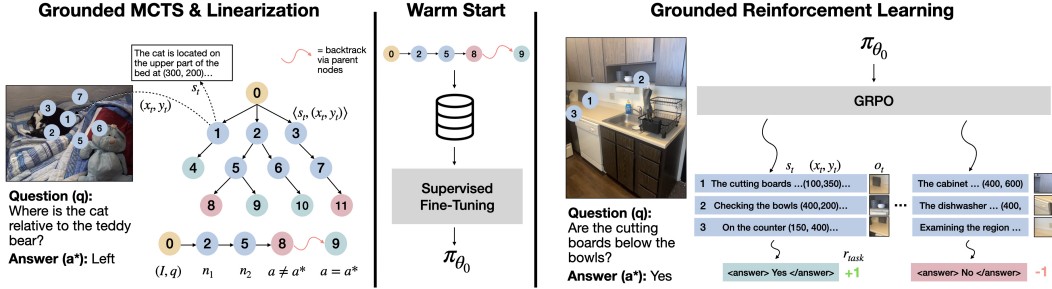

Figure 3: **Overview of the ViGoRL approach.** (Left) We use MCTS with a teacher model to generate reasoning chains grounded in specific image regions. (Middle) These reasoning trees are linearized and used for supervised fine-tuning (SFT) to train a base model. (Right) We apply GRPO with an outcome-based reward to further refine the grounded reasoning.

### 4.2 Warm-Start Data Generation via MCTS

**MCTS with Visual Grounding.** We employ MCTS to generate grounded reasoning traces, where each node is a reasoning step $n_t = \langle s_t, (x_t, y_t) \rangle$, anchoring thought $s_t$ to image coordinates $(x_t, y_t)$ (Figure 3, left panel). At each iteration: (1) *Selection*: nodes are traversed using UCB, prioritizing high-value, under-explored paths; (2) *Expansion*: the VLM samples new grounded steps $\langle s_t, (x_t, y_t) \rangle$ for unexplored nodes, each referencing a distinct image region; (3) *Simulation*: rollouts are performed

by recursively generating visually grounded steps until a terminal answer is produced; (4) *Backpropagation*: a judge scores terminal nodes, with rewards propagated up the path. This process ensures efficient exploration of promising image regions and reasoning steps. We provide additional details on our MCTS procedure in Appendix A5.5.

**Why MCTS?** Even large models (Qwen2.5-VL-72B) deployed with standard prompting explore only 2–3 regions, set few subgoals, and never backtrack (Table 5). Purely human-curated traces are costly to collect at scale, while linear rollouts cannot enforce the iterative exploration and corrective loops we desire. Empirically, distillation from such linear rollouts without MCTS leads to degraded generalization, performing worse on out-of-distribution spatial tasks after GRPO training (Table 4). In contrast, MCTS lets us systematically search the space of grounded reasoning steps, trading off exploration breadth and depth, and cheaply generate thousands of richly annotated paths that exhibit behaviors like wide exploration, through early branching to cover diverse image regions, and backtracking, by abandoning failing branches and revisiting alternatives.

Moreover, since our reasoning steps are already defined as $(s_t, (x_t, y_t))$ tuples, they map naturally onto MCTS nodes and transitions, making it both conceptually clean and computationally efficient.

**Teacher-guided Search.** We employ a frozen, high-capacity teacher (Qwen2.5-VL-72B) to expand each node in the MCTS tree. At node $n$, we prompt the teacher to either (a) generate a new grounded reasoning step $s_n$ with coordinate, or (b) emit a candidate answer. We score leaf answers by correctness and backpropagate to guide tree expansion. From 1,500 prompts we derive $\sim$30k high-quality reasoning traces—a dataset orders of magnitude smaller than typical SFT corpora, but *densely* packed with exploration, verification, and backtracking behaviors.

**Linearization for SFT.** We then linearize selected root-to-leaf paths into two types of training examples: 1) **Direct chains:** successful trajectories leading to the correct answer with no detours, and 2) **Corrected chains:** trajectories where an initial rollout fails, triggers a "wait, that seems off" backtrack, and proceeds to the correct solution (Figure 3, left panel).

We denote the VLM finetuned on these MCTS chains as $\pi_{\theta_0}$ (Figure 3, middle). After fine-tuning, we commonly observe reasoning chains with dense visual subgoal setting, visual verification, broad region exploration, and visual backtracking. A representative trace can be seen in Appendix A9.2.

### 4.3 Reinforcement Learning with Spatially-Grounded Reasoning Steps

Although $\pi_{\theta_0}$ imitates high-quality traces, it does not reason *optimally* for new queries. We therefore apply Group Relative Policy Optimization (GRPO; 50) to directly maximize task reward while preserving fluency and grounding (Figure 3, right). More details can be found in Appendix A5.1.

**Reward Design.** We wish to incentivize reasoning traces that include explicit grounding coordinates, and reasoning traces that lead to the final answer. Our total reward is a weighted sum: $R(\mathcal{T}) = \lambda_{\text{fmt}} r_{\text{fmt}} + \lambda_{\text{task}} r_{\text{task}}$, where $r_{\text{fmt}}$ encourages valid and interpretable output format, and $r_{\text{task}}$ captures task-specific correctness. Importantly, along with checking outputs are formatted with correct `<think>` and `<answer>` tags, we award $+1$ $r_{\text{fmt}}$ only if all coordinate references are valid. We provide more details on task reward in Appendix A5.2.

#### 4.3.1 Multi-Turn Reinforcement Learning for Visual Feedback

Grounded reasoning chains generated by our MCTS procedure already push the model to *look* at many regions, but the visual encoder still processes the *same* globally resized image at every step. Fine-grained cues (small text, icons, object boundaries) are therefore blurred away, potentially limiting the benefit of additional reasoning if image region details cannot be perceived by the model. Inspired by how humans zoom in after selecting a candidate region, we let the model request a higher-resolution crop, $o_t$, after predicting a coordinate. This "interactive microscope" supplies fresh evidence at a detail level impossible to encode in the initial global view.

**Multi-turn warm start: from single-turn chains to dialogs.** To prepare the model for multi-turn rollouts, we convert single-step MCTS-derived reasoning traces from Section 4.2 into dialog-style traces. Given a linearized MCTS trace $\tau = [(s_1, \mathbf{p}_1), \ldots, (s_T, \mathbf{p}_T), a]$, we convert it into a *dialog*:

1. At turn $t$, the model generates a textual thought $s_{t+1}$ (tagged `<think>` ... `</think>`).
2. The model then emits `<tool_call>` {"name": "crop", "arguments": {"coordinate": $\mathbf{p}_t$}} `</tool_call>`, or the answer (tagged `<answer>` ... `</answer>`), then the round terminates.

3. If a function call, the environment responds with $o_t$, with tag `<observation>` containing a $w \times w$ crop centered at $\mathbf{p}_t$, resized to $r \times r$ pixels, followed by `</observation>` and the loop repeats.

We fine-tune the base model on these multi-turn traces to initialize it for multi-step GRPO with visual feedback. We then apply GRPO, allowing the model to roll out full multi-turn dialogs, with crop feedback upon tool call outputs.

**Reward Design for Multi-Turn RL.** Multi-turn settings introduce new failure modes: repeating coordinates, skipping tool use, or violating dialog structure. To mitigate these, we define a composite format reward: **(1) Grammar reward** $r_{\text{grammar}}$: 1 if the dialog obeys a strict tag automaton: `<think>` $\rightarrow$ `</think>` $\rightarrow$ `<tool_call>` $\rightarrow$ `</tool_call>` $\rightarrow$ `<observation>` $\rightarrow$ `</observation>` $\rightarrow$ repeat or `<answer>` $\rightarrow$ `</answer>`. The dialog must end with a complete `</answer>` and contain no malformed or out-of-order tags. **(2) Diversity bonus** $r_{\text{div}}$: +0.2 for each sufficiently distinct coordinate in tool calls ($\geq$ 10px from all previous), up to 4 times. Finally, this leads to the overall format reward: $r_{\text{fmt}} = r_{\text{grammar}} + r_{\text{div}}$. Additional multi-turn RL details can be found in Appendix A5.3.

# 5 Experiments

We evaluate our model, ViGoRL, on spatial reasoning and web grounding tasks, comparing against baseline and ablation variants to understand the contributions of grounding, MCTS-based warm-start supervision, and GRPO. Our results demonstrate significant gains in visual reasoning and web-based tasks when models are trained with our grounded reasoning training recipe. We investigate the following research questions:
① **RQ1**: How much does grounded reasoning help when evaluated on visual reasoning tasks?
② **RQ2**: How important is each component of ViGoRL?
③ **RQ3**: What visual reasoning behaviors are amplified by grounded reasoning?
④ **RQ4**: Is the grounded reasoning accurate and interpretable?

**Training Datasets.** For **spatial reasoning**, we use SAT-2 [48], sampling 32k training and 1k validation examples. The model is tasked with selecting the correct textual option, with randomized answer order to reduce position bias. For **web grounding**, we draw 12k ⟨screenshot, referring expression, box⟩ examples from OS-ATLAS [69] (4k each from mobile, web, and desktop), plus 1.5k warm start and 1.5k validation samples evenly split by domain. For **web action prediction**, we use ICAL [49], a dataset of 92 web navigation trajectories. We remove chain-of-thought and textual set-of-marks annotations to focus on visual grounding, training on ⟨image, instruction, action history⟩ to predict the correct next action at each step. For **visual search**, we curate 11k question-answer pairs over small objects from Segment Anything [30], using GPT-4o given an object mask and scripted filtering to generate fine-grained ⟨image, question, choices, answer⟩ tuples. Each question targets a uniquely identifiable small object (<0.1% of image) and tests visual discrimination (e.g., color, material) and relation questions between object pairs.

## 5.1 RQ1: How much does grounded reasoning help on visual reasoning tasks?

**Baselines.** We compare against the following baselines:

1. *Method Comparison:* We compare ViGoRL to baselines utilizing the same training data and base model (Qwen-2.5-VL) as used in our method, but differing in their training recipe: 1) **SFT-direct:** Supervised fine-tuning on our trajectory dataset using final answers only. 2) **Vanilla GRPO:** GRPO applied to the base model with standard rewards for `<think>` and `<answer>` formatting, and answer correctness, similar to previous work [39, 40]. Our model and all baselines decode with a temperature of 0.5 during evaluation.

2. *General Proprietary and Open-Source Models:* General-purpose vision-language models with closed-weights accessible through APIs [1] and open-weight models [37, 11, 12, 62, 5, 58].

3. *VLM Tool-Using Pipelines:* We compare against prompt-based models that explicitly decompose visual tasks into sequences of subgoals or intermediate steps [66, 22, 26, 87, 75, 65].

4. *Web-Grounding Models:* Models trained to specialize in web grounding tasks, through large-scale supervised finetuning and instruction tuning on curated web data [10, 24, 69, 36, 19, 45], reinforcement learning of chain of thought with outcome reward [40], or human-in-the-loop action and chain of thought annotations collected from live webpages [49].

### 5.1.1 Visual Reasoning Evaluations.

**Spatial Reasoning.** We evaluate on SAT-2 [48] validation (4,000 questions across 5 categories), BLINK [15] (depth ordering, multi-view reasoning, spatial relationships), and ROBOSPATIAL-configuration and compatibility split (228 questions on real-world RGBD scenes) [53]. These benchmarks test generalization to novel environments, objects, and language configurations. Accuracy is measured via multiple-choice answer matching. We use a max side length of 1260 pixels while keeping aspect ratio for evaluation.

**GUI Understanding.** For grounding evaluation, we use ScreenSpot v2 [69, 10] (single-step localization), and ScreenSpot Pro [33] (high-resolution professional environments). To test small element grounding in low resolution, we additionally evaluate on ScreenSpot Pro with images downsampled to a lower resolution of 1920x1920, which we call ScreenSpot-Pro-LR (results in Appendix Table A6). Performance is measured by checking if predicted coordinates fall within target element bounding boxes. We also test on VisualWebArena [31] live web evaluation, using the visual-only configuration where models receive set-of-marks annotated webpage screenshots without HTML or text inputs. Task success is determined automatically by checking for specific criteria in the agent trajectory (see Koh et al. [31] for details). We use a resize image to have maximum pixel size of 3600x3600.

**Visual Search.** The visual search benchmark V*Bench tests fine-grained visual understanding using 191 high-resolution images from the SA-1B dataset (avg. 2246×1582). It includes 115 attribute recognition samples (e.g., color, material) and 76 spatial relationship samples, evaluating models' ability to analyze detailed object properties and relative positions in complex scenes. We use a resize image to have maximum pixel size of 3600x3600.

### 5.1.2 Results

Table 1: Performance (mean ± 95% CI) on ScreenSpot and VisualWebArena. Multi-t = multi-turn RL with visual feedback.

| Model | ScreenSpot -V2 | ScreenSpot -Pro | VWA (Vision Only) |
|---|---|---|---|
| *Proprietary Models* | | | |
| GPT-4o | 18.1 | 0.8 | 19.8 (w/ text) |
| Claude Comp. Use | - | 17.1 | – |
| *Open-source Models* | | | |
| Qwen2-VL-7B | - | 1.6 | 2.9 (w/ text) |
| Kimi-VL-16B-MoE | 92.8 | 34.5 | – |
| *Web Grounding Models* | | | |
| SeeClick | 55.1 | 1.1 | – |
| CogAgent-18B | - | 7.7 | – |
| OS-Atlas-4B | 71.9 | 3.7 | – |
| OS-Atlas-7B | 84.1 | 18.9 | – |
| ShowUI-2B | - | 7.7 | – |
| UGround-7B | - | 16.5 | – |
| UGround-V1-7B | – | 31.1 | – |
| UI-TARS-2B | 84.7 | 27.7 | – |
| UI-R1-3B | – | 17.8 | – |
| ICAL-7B | – | - | 8.2 (w/ text) |
| *Method Comparison* | | | |
| Qwen2.5-VL-3B | 68.4 (±2.6) | 23.9 | 4.2 (±1.3) |
| + SFT direct | 80.6 (±2.2) | 25.0 (±2.1) | 4.5 (±1.4) |
| + Vanilla GRPO | 84.4 (±2.0) | 29.0 (±2.2) | 4.8 (±1.4) |
| **ViGoRL-3b (Ours)** | **86.5** (±1.9) | **31.1** (±2.3) | **6.4** (±1.5) |
| **Multi-t ViGoRL-3b (Ours)** | 86.1 (±1.9) | 32.3 (±2.4) | - |
| Qwen2.5-VL-7B | 73.6 (±2.4) | 29.0 | 5.5 (±1.5) |
| **ViGoRL-7b (Ours)** | **91.0** (±1.6) | **33.1** (±2.3) | **11.2** (±2.1) |

Table 2: Accuracy (mean ± 95% CI) for spatial reasoning.

| Model | SAT-2 Val | BLINK | RoboSpatial |
|---|---|---|---|
| *Proprietary Models* | | | |
| GPT-4o | – | 60.0 | 76.2 |
| GPT-4 Turbo | – | 54.6 | – |
| Claude 3 Opus | – | 44.1 | – |
| Gemini Pro 1.0 | – | 45.2 | – |
| *General Open-source Models* | | | |
| LLaVA v1.6 34B | – | 46.8 | – |
| instructBLIP 13B | – | 42.2 | – |
| Molmo | – | - | 67.1 |
| *Method Comparison* | | | |
| Qwen2.5VL-3B | 46.1 (±1.5) | 44.4 (±2.3) | 54.4 (±6.5) |
| + SFT direct | 58.3 (±1.5) | 46.4 (±2.3) | 62.3 (±6.3) |
| + Vanilla GRPO | 50.0 (±1.6) | 46.5 (±2.3) | **69.7** (±6.0) |
| **ViGoRL-3b (Ours)** | **62.9** (±1.5) | **48.5** (±2.3) | 67.1 (±6.1) |
| Qwen2.5-VL-7B | 52.6 (±1.6) | 52.9 (±2.3) | 46.7 (±9.5) |
| **ViGoRL-7b (Ours)** | **67.5** (±1.5) | **54.1** (±2.3) | **76.4** (±7.5) |

**Grounded reasoning improves spatial accuracy.** As shown in Table 2, on a variety of spatial understanding benchmarks, ViGoRL significantly outperforms baselines without grounded reasoning. On SAT-2 test set, ViGoRL-3B achieves 62.9% accuracy—an improvement of +16.8 points over the base model and +12.9 points over Vanilla GRPO. Similar trends hold on the out of distribution benchmarks of RoboSpatial (67.1%) and BLINK (48.5%), with ViGoRL-3B outperforming SFT Direct by 2.3% and 4.8% on BLINK and RoboSpatial, respectively, and vanilla GRPO by 2.0% on BLINK. At the 7B scale, ViGoRL-7B further improves SAT-2 accuracy to 67.5%, demonstrating the

method's scalability and effectiveness across model sizes. We observe the same trend in BLINK and RoboSpatial, with improved accuracies of 54.1% and 76.4%, respectively. Importantly, these gains come without sacrificing general visual-language capabilities, as our method maintains performance on standard out-of-distribution VLM benchmarks (Table A7).

**Grounded reasoning helps localize complex web elements.** We evaluate performance on vision-only web interfaces (Table 1), where models must resolve ambiguous UI instructions via grounded image understanding. ViGoRL-3B outperforms both the base model, direct answer SFT, and vanilla GRPO across all tasks. For instance, on ScreenSpot-Pro, requiring grounding of small elements in high resolution webpage images, accuracy improves 21.8 points over the base model, 6.1 points over SFT direct, and 2.1 points over vanilla GRPO. ViGoRL-7B achieves 91.0% on ScreenSpot-V2 and 33.1% on ScreenSpot-Pro, outperforming open-source VLMs of comparable size, including OS-Atlas and UGround variants, which finetune on order of magnitudes more GUI grounding examples (up to 13M training samples).

**ViGoRL improves accuracy on live visual web evaluation.** On VisualWebArena, which requires live interaction with web pages using only set-of-marks image inputs (no access to HTML or underlying text), ViGoRL outperforms direct SFT and vanilla GRPO. Despite relying solely on images, ViGoRL surpasses the previous state-of-the-art for the same model size, ICAL [49], by 3.0%—even though ICAL has access to textual set-of-marks inputs derived from HTML.

Table 3: Accuracy (mean ± 95% CI) for visual search on V*Bench.

| Model Name | V*Bench |
| --- | --- |
| *Proprietary Models* | |
| Gemini-Pro | 48.2 |
| GPT-4V | 55.0 |
| GPT-4o | 66.0 |
| *VLM Tool-Using Pipelines* | |
| VisProg | 41.4 |
| VisualChatGPT | 37.6 |
| MM-React | 41.4 |
| Sketchpad-GPT-4o | 80.3 |
| IVM-Enhanced GPT-4V | 81.2 |
| *Open-source Models* | |
| LLaVA-1.5-7B | 48.7 |
| LLaVA-1.6-13B | 61.8 |
| SEAL | 74.8 |
| Qwen2.5-7B-VL | 78.0 (±3.0) |
| **Multi-t ViGoRL-7B** | **86.4** (±2.7) |
| Qwen2.5-3B-VL | 74.2 (±3.1) |
| ViGoRL-3B (Ours) | 79.1 (±3.0) |
| **Multi-t ViGoRL-3B** | **81.2** (±2.8) |
| w/o diversity reward | 78.0 (±3.0) |
| w/ bounding box outputs | **81.2** (±2.8) |

**Multi-turn RL with visual feedback improves visual search and small element detection.** On V*Bench, ViGoRL-7B significantly outperforms both proprietary models and open-source tool-using pipelines. As shown in Table 3, our method reaches 86.4%, surpassing proprietary VLMs like GPT-4o (66.0%) and Gemini-Pro (48.2%), and even advanced tool-using pipelines like VisProg (41.4%) Sketchpad-GPT-4o (80.3%).

Our model with multi-turn RL that incorporates zoomed-in visual feedback additionally shows significant improvements in small element grounding tasks. It achieves 32.3% accuracy on ScreenSpot-Pro, a 1.2 percentage point improvement over our best non-visual feedback model variant (31.1%) (Table 1). In low-resolution environments, the zooming capability delivers even more substantial relative gains, outperforming our best non-visual feedback method by 2.4% (Table A6). These gains illustrate that our approach enables more robust and precise visual reasoning through structured visual grounding and dynamic visual feedback interactions.

## 5.2 RQ2: Ablation Studies

Table 4: Ablation results on SAT-2 Val and BLINK. Top-1 accuracy (±95% CI) with relative change. *Model never produced correct formatting. Gnd = Explicit grounding in the reasoning steps. SFT = SFT Direct pretraining. Distill = warm-start teacher distillation.

| Variant | GRPO | MCTS | Gnd | Distill | SAT-2 | BLINK |
| --- | --- | --- | --- | --- | --- | --- |
| Full (ours) | ✓ | ✓ | ✓ | ✓ | 62.93 (±1.50) | 48.50 (±2.33) |
| – GRPO | | ✓ | ✓ | ✓ | 58.83 (±1.53) (–4.10%) | 44.97 (±2.32) (–3.53%) |
| – MCTS | ✓ | | ✓ | | N/A* | N/A* |
| – Grounded | ✓ | ✓ | | ✓ | 58.69 (±1.53) (–4.28%) | 45.44 (±2.32) (–3.06%) |
| Distill w/o MCTS | ✓ | | ✓ | ✓ | 63.28 (±1.49) (+0.35%) | 46.18 (±2.32) (–2.32%) |
| + SFT direct pre-train | ✓ | ✓ | ✓ | ✓ | 63.26 (±1.49) (+0.33%) | 48.56 (±2.33) (+0.06%) |

To understand which components drive performance, we conduct targeted ablations (Table 4):

**1. GRPO remains important.** Removing GRPO reduces performance by 4.1 points on SAT-2 and 3.5 points on BLINK, confirming the importance of RL refinement.

**2. MCTS-generated warm-start is essential.** Without structured traces from MCTS, the model fails to emit valid outputs, underscoring the need for scaffolded learning signals.

**3. Explicit grounding helps.** Running our same method without introducing explicit grounding in the warm start data resulted in a 3.4 point drop on BLINK, even with the same MCTS warm start and GRPO recipe.

**4. Teacher distillation without MCTS preserves in-distribution performance but degrades out-of-distribution generalization.** When using warm-start data from successful teacher linear rollouts (which contain grounded reasoning steps but no search), SAT-2 validation accuracy remains nearly unchanged (63.26% vs. 62.93%). However, performance on the more challenging out-of-distribution BLINK benchmark drops significantly by 2.32%.

**5. SFT direct pretraining adds little.** Adding an additional SFT stage to directly predict the answer before warm start training and GRPO yields marginal gains (<1 point), reinforcing the idea that grounded thought processes drive improvement.

**6. Pointing is sufficient for visual search.** As shown in Table 3, we observe no difference in V*Bench accuracy when switching to training ViGoRL to output variable size bounding box crops as opposed to the fixed-size point cropping used in ViGoRL. Fixed crops usually provide enough context and resolution for most cases, and ViGoRL can handle larger regions through multiple overlapping crops that together capture necessary information. Adaptive cropping could be more efficient but is limited by imprecise bounding boxes and inconsistent aspect ratios.

### 5.3 RQ3: What visual reasoning behaviors are amplified by grounded reasoning?

Following previous work on behavioral coding in language models [17], we code VLM behavior when tasked with the SAT-2 spatial reasoning benchmark, which requires synthesizing evidence across multiple image regions, in two scenarios: (1) Zero-shot prompting with "think step-by-step" instructions without explicit grounding, and (2) RL-tuned models including vanillo GRPO, ViGoRL, and an ablated version without explicit grounding incentives. We quantify reasoning behaviors on 300 representative SAT-2

Table 5: Average visual behaviors per example.

| Model | Regions Explored | Grounded Subgoals | Visually Verify | Backtrack | Acc. |
|---|---|---|---|---|---|
| *Qwen2.5-VL-72B (Zero-Shot)* | | | | | |
| Standard CoT | 2.3 | 0.27 | 0.27 | 0.00 | 0.65 |
| *Qwen2.5-VL-3B (Zero-Shot)* | | | | | |
| Standard CoT | 1.44 | 0.07 | 0.14 | 0.00 | 0.38 |
| *Qwen2.5-VL-3B (RL-tuned)* | | | | | |
| Vanilla GRPO | 1.8 | 0.00 | 0.17 | 0.00 | 0.48 |
| **Ours** | **3.5** | **1.1** | **0.39** | **0.47** | **0.64** |
| w/o grounding | 1.7 | 0.02 | 0.27 | 0.02 | 0.58 |

samples using GPT-4o evaluation, measuring visual regions explored, subgoal setting, verification, and backtracking. Additional study details can be found in Appendix A6.

**Findings.** As shown in Table 5, explicit grounding substantially amplifies visual reasoning behaviors. ViGoRL explores more visual regions (3.5 vs. 1.44/1.8 in zero-shot/vanilla GRPO) and demonstrates dramatic increases in grounded subgoals (1.1 vs. 0.07, 15× higher) and visual verification (0.39 vs. 0.14, 3× higher). It uniquely develops visual backtracking behavior (0.47) absent in all baselines. These enhanced behaviors enable our 3B parameter model to achieve accuracy (0.64) comparable to the 72B model (0.65). The ablation study confirms these benefits stem specifically from explicit grounding incentives, as removing them causes substantial regression in all measured behaviors, particularly regions explored (1.7 vs. 3.5) and grounded subgoals (0.02 vs. 1.1).

### 5.4 RQ4: Human Evaluation Shows Accuracy and Helpfulness of Grounded Reasoning

To assess our model's grounded reasoning traces, we conducted a human study evaluating whether predicted coordinates (1) correctly referred to the intended image region, and (2) helped participants understand the associated reasoning step (details are shown in Appendix A7).

**Findings.** As shown in Figure 4, 72.8% of predictions were judged as accurately referring to the described region (95% CI: [66.8, 78.7], N=20). On a 5-point Likert scale, participants rated the helpfulness of the highlighted region at 3.35 on average (95% CI: [3.03, 3.68], N=10). Helpfulness increased substantially when the prediction was correct (3.81; 95% CI: [3.51, 4.10]) and dropped when incorrect (2.26; 95% CI: [1.57, 2.94]).

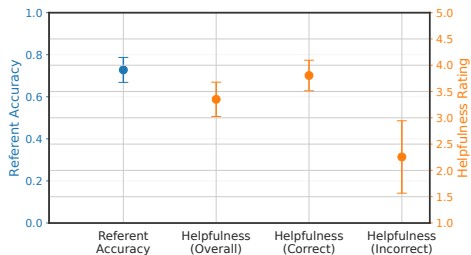

Figure 4: Human evaluation of grounded reasoning. Participants judged the grounded predictions as both accurate and helpful when correct.

*These findings indicate that accurate spatial grounding meaningfully improves human interpretability and usefulness of the model's reasoning process*, indicating that improvements in accuracy of reasoning step grounding can also improve human interpretability.

**Acknowledgments.** This material is based upon work supported by National Science Foundation grants GRF DGE1745016 & DGE2140739 (GS), ONR award N00014-23-1-2415, AFOSR Grant FA9550-23-1-0257, and DARPA No. HR00112490375 from the U.S. DARPA Friction for Accountability in Conversational Transactions (FACT) program. Any opinions, findings and conclusions or recommendations expressed in this material are those of the authors and do not necessarily reflect the views of the United States Army, the National Science Foundation, or the United States Air Force.

This research project has benefitted from the Microsoft Accelerate Foundation Models Research (AFMR) grant program through which leading foundation models hosted by Microsoft Azure along with access to Azure credits were provided to conduct the research.

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

# 6 Conclusion and Discussion

Why does visual grounding help? Our findings suggest that spatially anchoring each reasoning step forces the model to engage in a more structured, human-aligned form of cognition. ViGoRL learns to iteratively reference, inspect, and verify content in specific visual regions – amplifying cognitive behaviors such as subgoal formulation, visual verification, and backtracking.

This model architecture mirrors insights from cognitive science: humans rely on spatial attention and visual routines to decompose complex problems into manageable, perceptually grounded steps [61, 6]. Grounding serves not merely to reduce computational load (as human spatial attention is often characterized), but to scaffold reasoning with external visual structure – effectively using the content of the world as part of the thinking process [77, 23]. We observe similar benefits in models: spatial grounding enables better generalization, especially in out-of-distribution settings, and improves interpretability by making intermediate steps physically referable.

Rather than treating grounding as a visualization tool or auxiliary supervision signal, our results argue for it as a central architectural and algorithmic principle. By training models to reason with deictic reference – pointing, zooming, verifying – future systems may better reflect the iterative, grounded strategies that underlie human problem-solving. This opens up promising directions for building agents that not only reason effectively, but in ways that are queryable, adaptable, and aligned with perceptual experience.

# A1 Appendix Overview

The structure of this Appendix is as follows:

- Section A2 contains limitations and future work.
- Section A3 contains broader impacts.
- Section A4 contains additional discussion and connection to cognitive science.
- Section A5 contains additional methods details.
- Section A6 contains additional details on the model behavioral analysis.
- Section A7 contains additional details on the human evaluation.
- Section A8 contains additional experimental results.
- Section A9 contains example model outputs.

# A2 Limitations and Future Work

While our approach demonstrates strong improvements in visual reasoning performance, several limitations remain and open avenues for future research:

**Intermediate Reward.**   Our current reinforcement learning setup provides rewards only at the final answer level. In spatial reasoning tasks, this sometimes results in reward hacking, where the model receives positive reward despite generating partially incorrect or underspecified reasoning. Although our human evaluation confirms that the majority of grounding predictions are accurate and helpful, there remains measurable room for improvement. We hypothesize that introducing *dense intermediate rewards*—for both correct reasoning steps and accurate grounding—could mitigate these issues and improve alignment between grounding, reasoning, and reward. We leave this exploration to future work.

**Expanded Tool Use and Adaptive Control.**   While our model can point, crop, and zoom into visual regions, other forms of tool-augmented perception (e.g., highlighting, region comparison, or 3D navigation) may further support compositional reasoning. Future work could explore mechanisms that learn to balance visual actions dynamically.

**Learning When and How Much to Reason.**   We observe that the model often generates long reasoning chains, even for relatively simple questions. This inefficiency echoes known challenges in language-only chain-of-thought models [64, 32], where reasoning is applied uniformly regardless of

task complexity. A promising direction for future work is to develop methods that help models learn *when* to reason and *how much* reasoning is necessary, adapting the depth of their chain-of-thought to the demands of the task.

**Interpreting Attention Patterns.** Although our behavioral and human studies demonstrate that explicit grounding amplifies structured visual behaviors and improves interpretability, the mechanisms through which grounding improves reasoning remain underexplored. Future work should examine how explicit grounding influences a model's *internal* attention dynamics and whether these patterns resemble human gaze behaviors. Further comparisons to human cognitive strategies—such as visual routines [61], task-directed attention [77], and deictic planning [6]—could yield deeper insights into both model behavior and human cognition.

## A3   Broader Impacts

**Positive Societal Impacts.** Grounded reasoning improves model transparency by producing interpretable chains of thought that reference specific image regions. This interpretability is especially valuable in high-stakes domains such as medical imaging, accessibility tools, assistive robotics, and scientific image analysis, where understanding the basis for model decisions is crucial. Additionally, our approach may enhance human-AI collaboration by making the model's internal decision process legible and actionable for human users. Because the model reasons over localized visual subproblems, it can better align with human intuition, which may foster trust and oversight.

**Potential Negative Impacts.** Despite its benefits, the ability to produce detailed visual reasoning traces could be misused for surveillance or automated profiling. A grounded model that can isolate visual elements, track reasoning over time, and justify actions may inadvertently enable more fine-grained tracking of individuals or objects in sensitive contexts. Moreover, models that appear interpretable may be perceived as more reliable than they truly are; if the reasoning trace is superficially plausible but incorrect, users may over-trust the model's output. This risk is particularly relevant in domains where visual ambiguity or annotation bias affects ground truth.

Our approach also assumes the availability of visual data and may propagate or amplify dataset biases, especially in domains where certain object types, environments, or affordances are overrepresented. Spatial grounding mechanisms may lock the model's attention onto biased or stereotyped regions of the image, reinforcing existing disparities unless care is taken in dataset construction and evaluation.

**Mitigation Strategies.** To mitigate these risks, future work could incorporate uncertainty estimation or confidence calibration into the reasoning trace, explicitly marking when the model is uncertain about a grounding decision. It is also important to develop tools that allow users to audit the reasoning trace and provide corrections. In addition, efforts to diversify training data and evaluate model behavior across demographic and situational axes are essential for fair deployment. Finally, deployment in sensitive settings should involve human-in-the-loop oversight, particularly when grounded outputs inform consequential decisions.

## A4   Additional Discussion

Our results show that explicitly grounding reasoning steps in spatial coordinates substantially improves performance across visual reasoning benchmarks. To better understand why, we draw from both the recent machine learning literature and long-standing cognitive science theories that point to grounding as a core mechanism in human and artificial reasoning.

**Grounding as a Cognitive Scaffold.** Classic studies in visual cognition suggest that humans reason about complex scenes by sequentially directing attention to spatially localized regions in service of goal-driven subproblems. Yarbus [77] demonstrated that eye movement patterns depend heavily on task demands, suggesting a deep link between overt visual attention and internal reasoning. Ullman [61] introduced the concept of *visual routines*, showing that reasoning over a scene involves decomposable, spatially localized operations. Ballard et al. [6] formalized this via *deictic codes*, where visual fixations and gestures act as pointers binding abstract variables to perceptual content. These

mechanisms reduce *cognitive load* (as opposed to raw computational load), support compositionality, and facilitate subgoal execution by grounding symbolic reasoning in visual input.

**Grounding Reduces Hallucination and Enhances Generalization.** In machine learning, recent studies demonstrate that grounding in visual regions curbs hallucination and promotes generalization. Ghosh et al. [18] show that large VLMs hallucinate less and reason more accurately when first prompted to generate grounded visual descriptions. Qi et al. [44] propose a chain-of-manipulations method where VLMs perform spatially localized actions (e.g., zoom, crop, verify), achieving stronger generalization and interpretability. These findings echo Treisman's seminal Feature Integration Theory [60], which argues that spatial attention is essential for integrating visual features, and Harnad's symbol grounding problem [23], which proposes that abstract reasoning is only meaningful when connected to perceptual representations.

**Object-centric and Compositional Inductive Biases.** Explicit grounding may also encourage models to adopt object-centric and modular representations. Ding et al. [13] show that attention over learned object embeddings improves structured visual reasoning. Akula et al. [2] demonstrate that language-guided neural modules that condition on spatial cues improve the compositional generalization of the model. These results align with the cognitive perspective that humans naturally build scene representations around individual entities and their relations [6].

Overall, the convergence of cognitive science and ML literature suggests that explicit grounding does not merely reduce the search space, but acts as a structural inductive bias that enhances compositionality, verification, and goal-directed reasoning, core ingredients for generalization in both human and machine learners.

## A5 Additional Methodological Details

### A5.1 Group Relative Policy Optimization (GRPO)

Group Relative Policy Optimization (GRPO) [50] stabilizes policy learning from long-form trajectories by computing group-normalized advantages and applying clipped token-level PPO-style updates.

Specifically, for a group of $G$ trajectories $\mathcal{O} = \{\tau^{(i)}\}_{i=1}^{G}$ conditioned on input $x$, each trajectory $\tau^{(i)}$ has a scalar reward $r^{(i)} = R(\tau^{(i)})$. GRPO computes the centered advantage $\hat{A}^{(i)} = r^{(i)} - \bar{R}$, where $\bar{R} = \frac{1}{G}\sum_i r^{(i)}$ is the group mean.

Let $\tau_t^{(i)}$ be the $t$-th token of trajectory $\tau^{(i)}$. GRPO minimizes the following clipped surrogate loss:

$$\mathcal{L}_{\text{GRPO}}(\theta) = -\frac{1}{G}\sum_{i=1}^{G}\frac{1}{|\tau^{(i)}|}\sum_t \min\left[\rho_t^{(i)}\hat{A}^{(i)}, \text{clip}(\rho_t^{(i)}, 1-\varepsilon, 1+\varepsilon)\hat{A}^{(i)}\right] + \beta\,\text{KL}[\pi_\theta\|\pi_{\text{ref}}], \quad (1)$$

where $\rho_t^{(i)} = \frac{\pi_\theta(\tau_t^{(i)}|\tau_{<t}^{(i)},x)}{\pi_{\text{old}}(\tau_t^{(i)}|\tau_{<t}^{(i)},x)}$ is the importance weight, $\varepsilon = 0.2$ is the clipping parameter, and $\beta$ is the KL penalty coefficient. This approach stabilizes learning in long-horizon, multimodal reasoning settings.

### A5.2 RL Reward Functions

**Task Rewards.** We define reward functions specific to each benchmark:

- **Spatial Reasoning (SAT-2).** A binary reward: $r_{\text{task}} = 1$ if the predicted answer matches the ground truth, and 0 otherwise.
- **Web Grounding (OS-Atlas).** A binary reward: $r_{\text{task}} = 1$ if the predicted coordinate lies inside the annotated bounding box.
- **Web Action Prediction (ICAL).** A decomposed reward: $r_{\text{task}} = r_{\text{type}} + r_{\text{arg}}$, where $r_{\text{type}} = 0.5$ if the predicted action type matches, and $r_{\text{arg}} = 0.5$ if the predicted argument (e.g., DOM ID, string) matches.

### A5.3 Multi-turn Reinforcement Learning Details

We first apply supervised fine-tuning (SFT) on multi-turn traces. These are derived from the same MCTS chains used to train the single-turn model, but reformatted into multi-turn training data by:

(1) For each node, formulating the text in a think block followed by taking the coordinate and formulating it into a tool call. If it is a terminal node, a think block is added with any remaining text followed by the predicted answer in answer tags.

(2) Cropping around the coordinate to obtain the feedback image, and appending this image to the training data as a user turn in the sample to be used for SFT.

(3) Continuing (1) and (2) until terminal. We additionally include backtracking as described in Section 4.2.

For multi-turn scenarios, we apply GRPO over full dialog trajectories. Observation tokens are masked from the loss, ensuring gradients flow solely through the language model while retaining visual input to the encoder. We additionally mask samples not ending in an EOS token [79].

**Termination Enforcement.** During RL rollouts, if a dialog reaches $T_{\max} = 5$ turns without emitting an <answer> block, we append a soft prompt to the final assistant message:

```
<think> Please provide your response now </think>
```

This maintains structural fidelity and bounds rollout length.

**KL collapse**. While concurrent work observed occasional KL collapse in multi-turn RL when applied to base model with vision inputs [63, 91], we found that our initial warm start enabled stable training and allowed us to maintain a moderate KL coefficient (0.01).

**Diversity bonus.** We introduced the concept of a diversity turn bonus into our reward formulation in Section 4.3 of the main paper. In Figure A1, we show the response length over RL training with and without the diversity bonus. Without the diversity bonus, the model quickly converges to single turn outputs (green line), whereas the model avoids this and outputs >1 turn on average with the bonus reward (blue line).

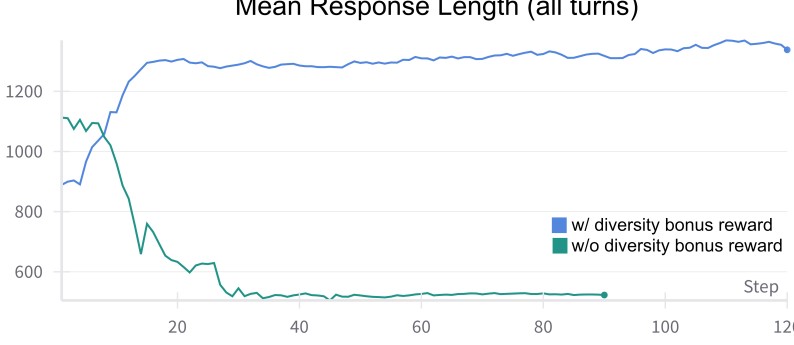

Figure A1: Response length with and without turn bonus. Without the bonus, the model converges to always taking a single turn (as also verified by examining model outputs), whereas the bonus enables the model to stabilize multi-turn.

### A5.4 Training Implementation Details

**General Setup.** Training is conducted on 8 A100 GPUs with Qwen2.5-VL models (3B and 7B). Supervised fine-tuning uses 3 epochs, while GRPO is applied for 500 rollout-update iterations. Evaluation uses decoding temperature of 0.5. We build on Llama-Factory [88] for SFT, and EasyR1 [89, 52] for GRPO.

**Training Hyperparameters Overview**  We provide all training hyperparameters used in SFT, and single- and multi-turn RL in Tables A1-A3.

Table A1: Supervised Fine-Tuning (SFT).

| Hyperparameter | Value |
| --- | --- |
| Epochs | 3 |
| Learning rate | 1e-6 |
| Weight decay | 0.01 |
| Warmup ratio | 0.03 |
| Batch size | 8 |
| Gradient accumulation | 4 |
| Effective batch size | 32 |
| Scheduler | Cosine |
| Precision | bf16 |
| Flash attention | fa2 |
| Freeze vision tower | True |
| Max sequence length | 8192 |
| Deepspeed config | ZeRO Stage 3 |

Table A2: GRPO Training.

| Hyperparameter | Value |
| --- | --- |
| Training steps | 500 |
| Learning rate | 1e-6 |
| Weight decay | 0.01 |
| Warmup ratio | 0.05 |
| Optimizer | AdamW (bf16) |
| Group size | 5 |
| KL coefficient | 0.01 |
| Clip ratio | 0.28 |
| Gradient clipping | Max norm 1.0 |
| Rollout batch size | 128 |
| Gradient step batch size | 32 |
| Rollout engine | vLLM |
| Max prompt length | 4096 |
| Max response length | 2048 |
| Top-$p$ | 0.99 |
| Temperature | 1.0 |
| Filter overlong | True |
| Freeze vision tower | True |
| Mixed precision | bf16 |

Table A3: Multi-turn GRPO.

| Hyperparameter | Value |
| --- | --- |
| Max prompt length | 4096 |
| Max response length (includes observations) | 4096 |
| Max generation per turn | 1024 |
| Max turns | 5 |
| Crop resize | 384x384 |
| Crop size | 100x100 |
| Learning rate | 1e-6 |
| KL coefficient | 0.01 |
| Weight decay | 0.01 |
| Warmup ratio | 0.05 |
| Clip ratio | 0.28 |
| Gradient clipping | Max norm 0.2 |
| Rollout batch size | 128 |
| Gradient step batch size | 64 |
| Training steps | 500 |
| Group size | 8 |
| Rollout engine | vLLM |
| Temperature | 1.0 |
| Top-$p$ | 0.99 |
| Mixed precision | bf16 |
| Freeze vision tower | True |

## A5.5  MCTS Implementation Details

To generate high-quality reasoning traces for fine-tuning, we use Monte Carlo Tree Search (MCTS) to explore possible sequences of reasoning steps over an image and question. Our MCTS procedure extends standard tree search with a VLM as the generative model for reasoning steps and a judge to evaluate final answers. We use the same reward as described in Section A5.2.

**Search Structure.**  Each node in the search tree corresponds to a reasoning step (a candidate `<think>` or `<answer>` block). The root node contains the input question. Nodes maintain the text of the current step, accumulated visited coordinates, a running estimate of expected reward, and visit count.

**Algorithm Phases.**  MCTS operates via the standard four-phase loop:

- **Selection:** Starting at the root, the search follows a path through children using the UCB policy: $Q + c\sqrt{\log N/n}$, where $Q$ is the average reward, $N$ is the parent's visit count, $n$ the child's, and $c = 2.0$ is the exploration constant. The search terminates when a node has unvisited children or reaches a terminal state.

- **Expansion:** At each expandable node, we generate up to three children using the VLM, each corresponding to a new thought or final answer. If a generated step includes a terminal `<answer>` block, it is marked as terminal. An example prompt for this step can be found in Listing A1.

- **Rollout:** For each new child, we simulate reasoning steps using the VLM until a final answer is reached or a rollout depth limit is exceeded. These simulated trajectories are not added to the tree. Each rollout receives a scalar reward from the judge model comparing the predicted and true answers.

- **Backpropagation:** Final 0/1 rewards are backpropagated up the tree along the visited path, updating the value estimate of each node using an incremental mean and incrementing visit counts. We use the same task reward as described in Section A5.2.

We include our MCTS hyperparameters in Table A4.

### A5.5.1 Linearization of Search Trees into Reasoning Chains

To convert MCTS-generated search trees into training data for supervised fine-tuning, we developed a structured linearization procedure that extracts diverse, grounded reasoning trajectories from the tree.

**Tree Traversal and Chain Extraction.** Each MCTS trace is stored as a tree, with nodes representing reasoning steps (`<think>`) or final answers (`<answer>`). From each trace, we recursively enumerate all root-to-leaf paths that include at least one terminal node. For each terminal node, we extract:

- **Correct rollouts**: Paths ending in a high-reward ($\geq 1.0$) final answer.
- **Incorrect rollouts**: Lower-reward paths used to generate synthetic backtracking examples.

If both correct and incorrect rollouts exist at a node, we concatenate the incorrect trace with a fixed backtracking phrase (e.g., "Wait, this seems off. Let's try something else.") followed by the corrected reasoning path, forming a complete trace with embedded revision behavior. All steps are wrapped in a single `<think>` block, and the final answer in a separate `<answer>` tag.

**Token Cleanup and Deduplication.** Each reasoning trace is cleaned to remove residual XML markers (`<think>`, `<answer>`) before joining, then wrapped again after concatenation. Chains are deduplicated across samples to prevent redundancy in training.

**MCTS accuracy.** As shown in Table A5, we observe improved top-1 accuracy using our MCTS procedure.

Table A4: MCTS hyperparameters for Web Grounding.

| Hyperparameter | Value |
|---|---|
| Model | Qwen2.5-VL-72B-Instruct |
| Simulations per input | 20 |
| Max tree depth | 10 |
| Rollouts per node | 2 |
| Children per expansion | 3 |
| $c_{puct}$ | 2.0 |
| Sampling temperature | 1.0 |
| Top-$p$ | 1.0 |
| Max tokens per node | 512 |
| Estimated time per sample | 21 minutes |
| Parallel processes | 10 |

## A6 Behavioral Analysis Protocol

To systematically evaluate reasoning behaviors in VLMs, we implement a behavioral annotation pipeline that categorizes model-generated chain-of-thought (CoT) traces using GPT-4o as an evaluator. This procedure is based on Gandhi et al. [17] and enables fine-grained analysis of emergent reasoning strategies across different model variants and training regimes.

**Sample Selection and Preprocessing.** We run the same 300 reasoning traces (obtained by randomly selecting examples from the SAT-2 validation set) per model condition where the model's final answer was verified correct (`judge_score = 1`). From each trace, we isolate the contents of the `<think>` block.

**Behavioral Categories.** We define four behavioral dimensions of interest:

- **Visual Verification:** Instances where the model confirms or checks a property of the visual scene (e.g., "Looking at the image, I can confirm...").
- **Backtracking:** Self-correction or re-interpretation of a previously described visual element.
- **Subgoal Setting:** Decomposition of the visual reasoning process into smaller steps across regions (e.g., "First I will check X, then Y...").
- **Visual Regions Explored:** Count of distinct visual regions explicitly referenced in the reasoning trace.

**Annotation via GPT-4o.**   For each trace, we construct four behavior-specific prompts and submit them to GPT-4o with temperature 0.0 and a max token limit of 256. Each prompt asks the model to identify and count instances of a specific behavior, outputting a numeric value between custom '<count>' tags. These counts are extracted and recorded as the behavioral profile for the trace. Our full prompts are displayed in Listing A2.

**Comparison Across Conditions.**   We apply the above process across multiple model variants (e.g., Our full model, Naive GRPO, Ablated grounded reasoning) and aggregate the behavior counts to compute the average number of reasoning behaviors per trace.

## A7   Human Evaluation Setup

To assess the interpretability and spatial accuracy of the model's grounded visual reasoning outputs, we conducted a structured human evaluation study using Prolific. The study was designed to evaluate whether the (x,y) coordinate output by the model accurately corresponded to the referenced region in the reasoning step, and whether this visual cue helped participants interpret the reasoning step. We obtained 80 samples randomly from the robospatial evaluation, as this evaluation contained real images without visual marks on the image.

**Data curation.**   We take 80 samples from our model run on Robospatial, and extract each sentence from the samples. To simplify the study for the participants and target step-level analysis, we filter out sentences that require context from the entire reasoning trace using GPT4o. For each reasoning step and coordinate, we draw a 100x100 pixel blue circle on the image centered at the coordinate location, and display the reasoning step text.

**Study Design.**   Each trial presented participants with a single image containing a blue circle annotation, along with a natural language sentence that included the phrase "[shown with blue circle]" (which replaced the (x,y) coordinate) to denote the region being referenced. Participants answered two questions per image:

1. **Accuracy (binary + unsure):** Participants were asked whether the blue circle overlapped with the region described by the sentence. Options were:
   - **Yes** — if any part of the described region was inside the blue circle.
   - **No** — if the region was entirely outside the circle.
   - **Unsure** — if the sentence was ambiguous or the region could not be clearly judged. Responses marked as "Unsure" were excluded from accuracy and clarity score calculations to avoid inflating or deflating agreement metrics with ambiguous judgments.

2. **Interpretability (Likert):** Participants rated how much the blue circle helped them understand the sentence reference on a 5-point Likert scale:
   - 1 — Strongly disagree
   - 2 — Disagree
   - 3 — Neutral
   - 4 — Agree
   - 5 — Strongly agree

**Interface and Instructions.**   Participants began the study by entering a participant ID. A detailed instruction panel introduced the task goals, decision criteria, and included two illustrative examples with images. These examples demonstrated both "Yes" and "No" cases for spatial accuracy, and clarified how to interpret the clarity rating.

**Participant Recruitment and Demographics.**   We recruited participants via the Prolific platform. A total of 20 participants completed the study, with all submissions manually reviewed and approved. The participant pool was demographically diverse: ages ranged from 21 to 71 (mean = 39.4, SD = 13.6), with balanced gender representation (11 female, 9 male). Participants were based in the United States or the United Kingdom, and all reported fluency in English. Participants completed the task in an average of 22 minutes, and were compensated at a rate of $12/hr.

## A8  Additional Experimental Results

**MCTS accuracy on OS-Atlas and SAT-2** We evaluate the accuracy of our MCTS procedure using Qwen2.5-VL-72B by measuring how often it reaches the correct answer, as determined by an oracle verifier. On 123 held-out samples from OS-Atlas (Table A5), MCTS achieves an accuracy of 82.1%, indicating strong search effectiveness when guided by a reliable verifier. On the SAT-2 benchmark, MCTS reaches 96% accuracy.

**ScreenSpot-Pro Low Resolution Results.** We provide results on ScreenSpot-Pro-LR in Table A6. This variant of ScreenSpot-Pro has downsampled images to a max resolution of 1920x1920. On ScreenSpot-Pro-LR, the base Qwen2.5-VL-3B model achieves only 1.96% accuracy, while SFT improves performance to 16.89%. Incorporating GRPO further increases accuracy to 20.30%, and our full ViGoRL achieves 21.32%. Notably, our multi-turn ViGoRL reaches 23.72%, demonstrating that iterative visual feedback is especially beneficial in challenging perceptual settings.

**Accuracy on Out-of-Distribution Benchmarks**. To ensure that our model does not lose foundational knowledge after our pipeline, we evaluate on some popular VQA benchmarks: MMMU [80], Real-WorldQA [70], and V* [66]. On MMMU and RealWorldQA, we evaluate our model's performance when prompted without thinking, thus performing an apples-to-apples comparison to the base model. On V*, we prompt our model to think to compare to the V* visual search paradigm. On all datasets, we find that ViGoRL either matches or exceeds the performance of the base model.

**Splitting versus combining training datasets.** To quantify the effect of combining data, we conducted an experiment combining spatial and web datasets during both warm-start and RL phases. The results, shown in Table A8, reveal:

- Web grounding: Small decrease (-0.2% on ScreenSpot-Pro, -0.3% on ScreenSpot-V2)

- Spatial reasoning: Improvement of +1.1% on SAT-2

- Both approaches significantly outperform vanilla GRPO (+12-15% SAT-2)

These findings indicate that combining data has small effects on relative performance gains from our method, with slight benefits for spatial reasoning at the cost of web grounding performance.

**Euclidean distance reward for grounding leads to lower accuracy.** As shown in Table A9, we tested a Euclidean distance reward, instead of binary reward, for grounding and found it significantly reduced performance: -2.2% on ScreenSpot-V2 and -0.7% on ScreenSpot-Pro (Table R3).

Web elements vary in size, so a distance-based reward unfairly penalizes valid clicks near the edges of large elements. For instance, clicking near the edge of an 800×60 navigation bar is functionally correct but would receive a low euclidean reward.

**Comparison to teacher Qwen2.5VL-72B model** We show zero-shot performance of our teacher model (Qwen2.5-VL-72B) in Table A10 below. Despite using only 1k examples for distillation before RL, ViGoRL-7B outperforms Qwen2.5-VL-72B by 6.0% on SAT-2 while being 10× smaller, and closes the gap across most benchmarks.

Table A5: Held-out performance of Qwen2.5-VL-72B on 123 samples in OS-Atlas validation and 958 samples in SAT-2 validation. * indicates MCTS Accuracy of the hold-out MCTS set used in the training pipeline.

| Model | Top-1 Accuracy | Top-3 Accuracy | MCTS Accuracy |
|---|---|---|---|
| OS-Atlas | 42.3% | 54.5% | 82.1% |
| SAT-2 | 61.48% | 76.30% | 96.0%* |

Table A6: Accuracy (mean with 95% confidence intervals) on ScreenSpot-Pro-LR benchmark.

| Model | ScreenSpot-Pro-LR |
|---|---|
| Qwen2.5-VL-3B Base | 1.96% (±0.68) |
| + SFT direct | 16.89% (±1.85) |
| + Vanilla GRPO | 20.30% (±2.24) |
| **ViGoRL-3b (Ours)** | 21.32% (±2.28) |
| **ViGoRL-3b (Multi-turn)** | **23.72%** (±2.39) |

Table A8: Accuracy (mean ± 95% CI) on ScreenSpot-Pro/V2 and SAT-2.

| Condition | ScreenSpot-V2/Pro | SAT-2 |
|---|---|---|
| Qwen2.5-3B-VL | 23.9 (±1.4) / 68.4 (±2.6) | 46.1 (±1.5) |
| Vanilla GRPO (seperate spatial or web) | 29.0 (±2.2) / 84.4 (±2.0) | 50.0 (±1.6) |
| ViGoRL-3B (seperate spatial or web) | 31.1 (±2.3) / 86.5 (±1.9) | 62.9 (±1.5) |
| ViGoRL-3B (combined spatial+web) | 30.9 (±2.3) / 86.2 (±2.0) | 64.0 (±1.5) |

Table A9: Euclidean vs. binary reward ablation. Accuracy (mean ± 95% CI) on ScreenSpot-V2 and ScreenSpot-Pro.

| Condition | ScreenSpot-V2 | ScreenSpot-Pro |
|---|---|---|
| Qwen2.5-3B-VL | 68.4% (±2.6) | 9.3% (±1.4) |
| ViGoRL-3B w/ binary bbox reward | 86.5% (±1.9) | 31.1% (±2.3) |
| ViGoRL-3B w/ euclidean reward | 84.3% (±2.0) | 30.4% (±2.3) |

Table A7: Accuracy (mean with 95% confidence intervals) on Out-of-Distribution benchmarks. * indicates that the model was prompted to think.

| Model | MMMU | RealWorldQA | V* Bench |
|---|---|---|---|
| Qwen2.5-VL-3B Base | 47.44% (±3.26) | 55.65% (±3.52) | 74.21% (6.22) |
| **ViGoRL-3b (Ours trained on SAT-2)** | 46.44% (±3.26) | 60.65% (±3.46) | 74.87% (±6.15)* |
| **ViGoRL-3b (Ours trained on UGround)** | 47.56% (±3.26) | 57.52% (±3.50) | 75.13% (±6.16)* |

# A9 Example Model Outputs

## A9.1 Failure Cases in Current VLMs

Despite high image-text alignment, models like Qwen2.5-VL-7B fail often produce abstract, shallow reasoning without examining many different image regions or describing content in visual depth, which often leads to errors. A representative trace is below:

```
<think> From the image, the twotiered metal cart is positioned closer to the modern TV stand in blue.
The cart is nearer to the TV stand, which is evident from its proximity and the angle at which it is seen.
Therefore, the twotiered metal cart is closer to the modern TV stand in blue. </think> <answer>twotiered
metal cart</answer>
```

## A9.2 Warm Start Reasoning Traces

Fine-tuning with MCTS-generated warm traces leads to structured and grounded reasoning. Example trace:

Table A10: Comparison to Qwen2.5-72B-VL. Accuracy (mean ± 95% CI) across benchmarks.

| Model | SAT-2 | BLINK | RoboSpatial | ScreenSpot-V2 | ScreenSpot-Pro |
|---|---|---|---|---|---|
| Qwen2.5-VL-72B | 61.5% (±1.5) | 59.5% (±2.3) | 55.7% (±6.5) | 88.1 (±1.8) | 34.2 (2.6±) |
| ViGoRL-7B | 67.5% (±1.5) | 54.1% (±2.3) | 76.4% (±7.5) | 91.0% (±1.6) | 33.1% (±2.3) |

Table A11: ViGoRL error analysis by type on SAT-2. Percent of failures by category.

| Error Type | Percentage | Description |
|---|---|---|
| Excessive Reasoning | 20% | The model overthinks the problem. |
| Incomplete Reasoning | 10% | The model truncates its reasoning process. |
| Hallucination | 10% | The model invents visual elements. |
| Challenging Visual QA | 30% | The question or image presents inherently subtle cues, causing reasoning errors. |
| Base Perception | 30% | The underlying model makes perceptual or conceptual grounding errors. |

```
<think>
I need to locate Candice Georgiadis in the list. I see the name Candice Georgiadis at (330, 440). Now I will
check for the 'Follow' button next to it. Next to Candice Georgiadis's name at (330, 440), I see a 'Follow'
button at (440, 440). I will verify if there are any other mentions of this name. I should check another area
to ensure there's no other 'Candice Georgiadis'. There's another sibling list and see the name Candice
Georgiadis at (660, 770) but no 'Follow' button is associated with this name. Having verified the name
'Candice Georgiadis' and the 'Follow' button, and ensuring there was no other mention of the name with a
follow button, I am confident the desired location is at the first identification at (440, 440).
Wait, this seems off. Let's try something else.
I need to locate Candice Georgiadis in the list. I see the name Candice Georgiadis at (330, 440). Now I
will check for the 'Follow' button next to it. Next to the name Candice Georgiadis at (330, 440), I see a
button at (365, 450) that looks like it says 'Follow'. I should check another name on the page for similar
formatting to confirm this is the correct 'Follow' button. To ensure that this is indeed the correct 'Follow'
button, I will check the name John Doe and its corresponding button at (230, 210). This button has the
same format as the one I observed for Candice Georgiadis at (365, 450). It reinforces that the button next
to Candice Georgiadis is the one I need. Lastly, to eliminate all doubt, I will examine a different region of
the image where the 'Follow' button for Fred Smith at (500, 600) is located. It aligns with the same design
as the others, confirming the earlier observations. Therefore, I can now finalize the answer for Candice
Georgiadis's 'Follow' button.
</think>
```

## A9.3 Additional Model Output Examples

We provide example outputs on RoboSpatial questions for two models: the vanilla GRPO model
trained on SAT-2 (Figure A2), and our proposed ViGoRL trained on SAT-2 (Figure A3).

## A9.4 Failure cases

We categorizes SAT-2 errors in Table A11. 60% arise from perceptual or task-level challenges, while
40% reflect residual reasoning flaws (e.g., overthinking, truncation, hallucination). We believe scaling
or denser rewards could address aspects of both, with further discussion in Section 6 and Appendix
A2.

**Example of overthinking in ViGoRL for simple question.**
**Question:** "Is there a pot in the image?"

**Image:** *Image of a stove with a large pot and pan on it.*

**Model response:**

```
<think>
To determine if there's a pot, I'll check the regions where pots typically appear. At (300, 600), there's
a pot-like object with a handle and a lid, suggesting it could be a pot. To confirm the presence of a pot,
I'll check another region. At (700, 1000), there's no pot-like object present, indicating that the previous
observation was accurate. Now, I'll verify if there are any other potential pots in the image. At (500, 800),
there's no pot-like object present, which supports the previous findings. Next, let's examine the area at
(400, 900) to ensure there's no hidden pot. There's still no pot-like object visible here, reinforcing the
previous observations. I'll now check the lower region at (600, 1200) to confirm there are no additional
pots. At (600, 1200), no pot-like object is visible here, further supporting the absence of a pot in the image.
Lastly, I'll check the upper region at (300, 200) to ensure there's no hidden pot. At (300, 200), no pot-like
object is visible here, confirming the pot is at (300, 600) in the image.
</think>
```

## Listing A1: MCTS prompt for web grounding.

```
You are a helpful assistant tasked with grounding an element on a web page. You should
    systematically reason through the problem step by step by checking and verifying
    relevant webpage regions, while grounding reasoning steps to specific (x, y) points
    in the image:\nEach reasoning step must be enclosed within '<think>' tags and
    reference exactly one specific coordinate (x, y):\n<think>\n{Single reasoning step
    with a grounded point} (x, y).\n</think>\nWhen ready to provide the final answer,
    enclose it within '<answer>' tags:\n<answer> (xf, yf) </answer>\nYour task is to
    help the user identify precise (x,y) coordinates of a described area/element/object
    based on a description.\n- Generate ONLY ONE reasoning step OR the final answer per
    response.\n- Regions are distinct, non-overlapping areas (e.g., quadrants like top-
    left, elements like tree/button, zones like background/foreground).\n- Each step
    should describe the region then evaluate it for its relevance to the task and to
    previous steps.\n- Never repeat coordinates from previous steps.\n- Begin by
    exploring diverse regions, even if they seem less likely, to ensure comprehensive
    coverage before narrowing down.\n- Prioritize broad coverage of diverse candidates
    before deciding.\n- Aim for accurate, representative points in the described area/
    element/object.\n- If unclear, infer based on likely context or purpose.\n- Verify
    each step by examining multiple possible solutions before selecting a final
    coordinate.\n- Format points as (x, y)
```

Listing A2: Prompts used to evaluate visual reasoning behaviors in chain-of-thought outputs.

```
Here is a chain-of-reasoning that a Language Model generated while analyzing an image.
    The chain-of-reasoning output from the model is:
'''
{reasoning}
'''
Evaluate whether the chain-of-reasoning contains any visual verification steps. A visual
    verification step is when the model confirms or checks something it sees in the
    image. Examples include: "I can see that the object is not a cat, but a dog", "The
    text confirms this visual aspect is correct", "I can verify this is indeed red", or
    "Looking at the image, I can confirm...". Count both explicit mentions of image
    regions and implicit verifications.
Count all instances where the model verifies information from the image and provide the
    count between the tags <count> </count>. If the chain-of-reasoning does not contain
    any visual verification steps, please provide a count of 0 as <count>0</count>.

---

Here is a chain-of-reasoning that a Language Model generated while analyzing an image.
    The chain-of-reasoning output from the model is:
'''
{reasoning}
'''
Evaluate whether the chain-of-reasoning contains any backtracking behavior, where the
    model changes its interpretation or corrects itself. Examples include: "At first I
    thought X, but looking more carefully I see it's actually Y", "I initially
    interpreted this as a circle, but it's actually an oval", "On second thought...", "
    Actually, I notice that...", or "Let me correct my earlier observation...".
Count all instances where the model revises its understanding and provide the count
    between the tags <count> </count>. If the chain-of-reasoning does not contain any
    backtracking behavior, please provide a count of 0 as <count>0</count>.

---

Here is a chain-of-reasoning that a Language Model generated while analyzing an image.
    The chain-of-reasoning output from the model is:
'''
{reasoning}
'''
Evaluate whether the chain-of-reasoning contains any visual subgoal setting, where the
    model breaks down the image analysis into smaller steps or focuses on different
    parts of the image in sequence. Examples include: "First, I'll examine this part,
    then I'll look at that object", "Let me check each element one by one", "I need to
    identify what's in this area", or any structured approach to analyzing different
    parts of the image.
Count all instances where the model sets up a plan or approach for analyzing the image
    and provide the count between the tags <count> </count>. If the chain-of-reasoning
    does not contain any visual subgoal setting, please provide a count of 0 as <count
    >0</count>.

---

Here is a chain-of-reasoning that a Language Model generated while analyzing an image.
    The chain-of-reasoning output from the model is:
'''
{reasoning}
'''
Count how many distinct visual regions or elements the model explicitly mentions
    examining in the image. Examples include: "I can see a dog in the corner", "There's
    text at the top of the image", "The object in the center appears to be...", "Looking
     at the left side...", or any reference to a specific part or element of the image
    that the model is analyzing.
Count all distinct visual regions or elements mentioned and provide the count between the
     tags <count> </count>. If the chain-of-reasoning does not mention any specific
    visual regions, please provide a count of 0 as <count>0</count>.
```

| Images | Question | Base Model (SFT Direct) | Naive GRPO Model |
|---|---|---|---|
| | Q. Can the table fit above the bed? Answer yes or no. Answer with the text of the option.. 

 GT Ans :- Yes | Thoughts: 

 Predicted Answer: No | Thoughts: From the image, the table is positioned to the right of the bed. The table top is level with the side of the bed, and the surface of the bed is a foot or two higher. Therefore, the table can fit above the bed. 

 Predicted Answer: Yes |
| | Q. Can the cactus fit in front of the litter box? Answer yes or no. Answer with the text of the option.. 

 GT Ans :- Yes | Thoughts: 

 Predicted Answer: No | Thoughts: Since the litters are next to the litter box, the cactus can be placed in front of the litter box because it is a long space for plants, and it can accommodate the height from the ground to the cactus. But if the plants are taller, the cactus will not fit in front of the litter box. 

 Predicted Answer: Yes |
| | Q. Can the pan fit above the fridge? Answer yes or no. Answer with the text of the option.. 

 GT Ans :- Yes | Thoughts: 

 Predicted Answer: No | Thoughts: From the image, the fridge appears to be open and wider than the stove, allowing the pan to be lifted above it. This suggests that the pan can fit above the fridge. Therefore, the answer is "yes". 

 Predicted Answer: Yes |

Figure A2: Example reasoning traces from vanilla GRPO on the RoboSpatial dataset, which does GRPO directly on the base model without warm start.

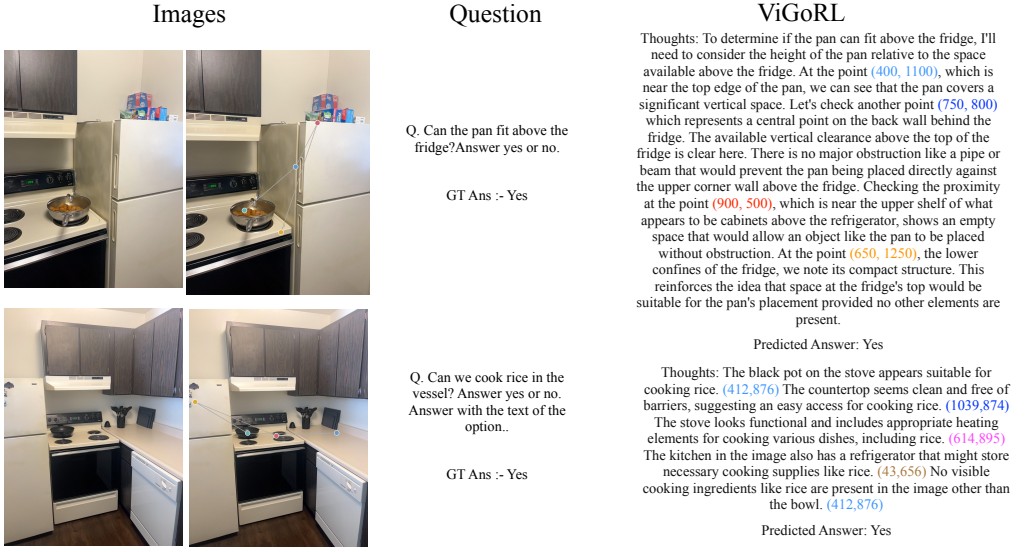

| Images | Question | ViGoRL |
|---|---|---|
| | Q. Can the pan fit above the fridge? Answer yes or no. 

 GT Ans :- Yes | Thoughts: To determine if the pan can fit above the fridge, I'll need to consider the height of the pan relative to the space available above the fridge. At the point (400, 1100), which is near the top edge of the pan, we can see that the pan covers a significant vertical space. Let's check another point (750, 800) which represents a central point on the back wall behind the fridge. The available vertical clearance above the top of the fridge is clear here. There is no major obstruction like a pipe or beam that would prevent the pan being placed directly against the upper corner wall above the fridge. Checking the proximity at the point (900, 500), which is near the upper shelf of what appears to be cabinets above the refrigerator, shows an empty space that would allow an object like the pan to be placed without obstruction. At the point (650, 1250), the lower confines of the fridge, we note its compact structure. This reinforces the idea that space at the fridge's top would be suitable for the pan's placement provided no other elements are present. 

 Predicted Answer: Yes |
| | Q. Can we cook rice in the vessel? Answer yes or no. Answer with the text of the option.. 

 GT Ans :- Yes | Thoughts: The black pot on the stove appears suitable for cooking rice. (412,876) The countertop seems clean and free of barriers, suggesting an easy access for cooking rice. (1039,874) The stove looks functional and includes appropriate heating elements for cooking various dishes, including rice. (614,895) The kitchen in the image also has a refrigerator that might store necessary cooking supplies like rice. (43,656) No visible cooking ingredients like rice are present in the image other than the bowl. (412,876) 

 Predicted Answer: Yes |

Figure A3: Example reasoning traces from ViGoRL on the RoboSpatial dataset.

