# OpenReview forum: "Grounded Reinforcement Learning for Visual Reasoning"
_NeurIPS.cc/2025/Conference — NeurIPS 2025 poster_

### Official Review · Reviewer_Dj1B · 2025-06-22

**Clarity:** 4
**Significance:** 4
**Originality:** 4
**Rating:** 4
**Confidence:** 4

**Summary:**

Summary

1.	The paper introduces ViGoRL, a vision-language model designed to enhance visual reasoning capabilities through reinforcement learning (RL).

2.	The motivation is to compose reasoning step explicitly anchored in the image coordinate, i.e. n = <s, (x,y)>. It can encourage the model to do more targeted and systematic cross-reference between text and image.

3.	It proposes a two-stage pipeline including warm-start supervised finetuning to produce structured reasoning steps with MCTS data generation, and reinforcement learning to refine grounding response with multi-turn visual signal.

4.	ViGoRL has been assessed across a range of visual reasoning benchmarks, including SAT-2 and BLINK for spatial reasoning tasks, VBench for web-based action prediction, and ScreenSpot and VisualWebArena for web-based grounding challenges. The model demonstrates superior performance compared to conventional supervised fine-tuning methods and RL baselines that lack explicit grounding components.

**Questions:**

Comment

1.	Since a teacher model (Qwen2.5-VL-72B) is used to distill grounded reasoning traces, it is required to provide to the zero-shot performance of the teacher model in Table I, Table II and Table III.

2.	An ablation study in terms of accuracy on reward design [line220-225] should be provided.

3.	The contribution of multi-turn visual feedback is unclear. In Table I, the enhancement of small element grounding (ScreenSpot-Pro) is not obvious and the degradation of Screenspot-V2 need to be further elaborated.

4.	Line 800, for the reward of Web Grounding, it is curious about why don’t use the euclidean distance between the predicted coordinates and center of bounding box.

5.	The crop size for multi-turn GRPO is fixed to 100x100. However, some objects such as the pendant light in Figure 1 is not with a regular shape. How to deal with such problem?

**Ethical Concerns:**

["NO or VERY MINOR ethics concerns only"]

**Final Justification:**

Most of my concerns have been addressed. This work is demonstrated as a SOTA work. It is helpful in the multimodal domain.

**Limitations:**

yes

**Quality:**

3

**Strengths And Weaknesses:**

Strength

•	Attaching grounding signal during reasoning stages is of novelty.

•	Presentation of this work is easy to follow.

•	Contribution of this work for visual grounding field is significant.


Weakness

•	SOTAs performance is not demonstrated in this work.

•	Out-of-distribution generalization should be further improved.

---

> ### Author Rebuttal · Authors · 2025-07-31
>
> Thank you for your thoughtful review. We were pleased that you recognized our work's novelty in attaching grounding signals during reasoning stages, appreciated our clear presentation, and found our contribution to the visual grounding field significant.
>
> We have addressed concerns through additional experiments and analysis. To summarize: (1) we conducted reward ablations, (2) we validated the contribution of multi-turn visual feedback on V\*Bench, (3) we clarify our state-of-the-art performance, (4) we clarify our OOD generalization, (5) we empirically validated our binary reward design over Euclidean distance, (6) compared adaptive vs. fixed cropping with experiments, and (7) we added teacher model comparisons.
>
> **We present our answers in detail below and hope we’ve addressed your concerns. Please let us know if anything remains unclear. We would be happy to provide further clarification.**
>
> > **Reviewer comment:** Reward ablation study should be provided.
>
> We thank the reviewer for this valuable suggestion. We have ablated all components of our reward formulations in Tables R1 and R2 below.
>
> Specifically:
> - Removing the multi-turn diversity bonus reward leads to a decrease of 3.2% on V\*bench. Appendix Figure A1 shows that removing this term leads to turn collapse.
> - Removing the format reward leads to a decrease of 0.6% on SAT-2.
> - Removing the accuracy reward leads to a decrease of 2.3% on SAT-2.
>
> We will include these ablations in Section 5.2 of the revised manuscript.
>
> ---
>
> ### Table R1: Multi-turn ablations
>
> | **Condition**                  | **V\*Bench**    |
> | ------------------------------ | --------------- |
> | **Multi-t ViGoRL-3B (Ours)** | **81.2 (±2.8)** |
> | └─ w/o diversity reward        | 78.0 (±3.0)     |
> | └─ w/o text thoughts           | 80.0 (±2.9)     |
> | └─ **w/ bounding box outputs**           | **81.2 (±2.8)**    |
>
> ---
>
> ### Table R2: Additional reward ablations
>
> | **Condition**          | **SAT-2**    |
> | ---------------------- | ------------ |
> | ViGoRL-3B (Ours)     | 62.9% (±1.5) |
> | └─ w/o format reward   |    62.3% ± (±1.5)          |
> | └─ w/o accuracy reward |      60.6% ± (±1.5)        |
>
> ---
>
> > **Reviewer comment:** The crop size for multi-turn GRPO is fixed. How to deal with irregular shapes?
>
> We conducted additional experiments where both MCTS warm-start and RL phases in ViGoRL use predicted bounding boxes (adaptive crops) instead of fixed-size point-based crops.
>
> **As shown in Table R1 above, we observe no difference in V\*Bench accuracy between the adaptive bounding box and fixed-point cropping approaches.** We attribute this to two main reasons:
>
> 1. **Sufficient context from fixed crops:** The default crop size is typically large enough to capture both the target region and its surrounding context at a high resolution for most V\*Bench examples.
>
> 2. **Multi-crop coverage of larger objects:** When the region of interest is too large for a single fixed crop, ViGoRL samples multiple overlapping crops in the vicinity, analogous to human multiple fixations, which suffices to infer the correct answer.
>
> While adaptive cropping could, in principle, be more efficient, we encountered practical limitations:
>
> * Bounding box predictions from Qwen are often imprecise.
> * Inconsistent aspect ratios from adaptive crops can introduce training variability.
>
> We agree this is a valuable design dimension and plan to further explore trade-offs in future work.
>
> > **Reviewer comment:** Why not use the euclidean distance between the predicted coordinates and center of bounding box?
>
> We thank the reviewer for the suggestion. **We tested the Euclidean distance reward and found it significantly reduced performance: -2.2% on ScreenSpot-V2 and -0.7% on ScreenSpot-Pro (Table R3).** We will include this in the revision.
>
> Web elements vary in size, so a distance-based reward unfairly penalizes valid clicks near the edges of large elements. For instance, clicking near the edge of an 800×60 navigation bar is functionally correct but would receive a low euclidean reward.
>
> ---
>
> ### Table R3: Euclidean vs. binary reward ablation
>
> | **Condition**                     | **ScreenSpot-V2** | **ScreenSpot-Pro** |
> | --------------------------------- | ----------------- | ----------------- |
> | Qwen2.5-3B-VL                     | 68.4% (±2.6)               | 9.3% (±1.4)               |
> | ViGoRL-3B w/ binary bbox reward | 86.5% (±1.9)   | 31.1% (±2.3)   |
> | ViGoRL-3B w/ euclidean reward   | 84.3% (±2.0)          | 30.4% (±2.3)          |
>
> ---
>
> > **Reviewer comment:** The contribution of multi-turn visual feedback is unclear.
>
> We find that multi-turn visual feedback is especially valuable when the model cannot perceive fine-grained image details in the initial input resolution. ScreenSpot-V2 has large, visible elements, so feedback offers no gain (–0.4%), while ScreenSpot-Pro features smaller, high-resolution targets, where feedback improves performance (+1.2%).
>
> **To validate this claim further, we evaluated ViGoRL on V\*Bench in Table R4, which requires identifying small details in ultra high-resolution images. Multi-turn ViGoRL outperforms the single-turn variant by 2.1%**, due to better disambiguating details via the zoomed-in crops. This reinforces that visual feedback is crucial when the global image tokenization blurs away fine-grained cues.
>
> ---
>
> ### Table R4: Accuracy for visual search on V\*Bench
>
> | **Model Name**                 | **V\*Bench**    |
> | ------------------------------ | --------------- |
> | *Proprietary Models*           |                 |
> | Gemini-Pro                     | 48.2            |
> | GPT-4V                         | 55.0            |
> | GPT-4o                         | 66.0            |
> | *VLM Tool-Using Pipelines*     |                 |
> | VisProg                        | 41.4            |
> | VisualChatGPT                  | 37.6            |
> | MM-React                       | 41.4            |
> | Sketchpad-GPT-4o               | 80.3            |
> | IVM-Enhanced GPT-4V            | 81.2            |
> | *Open-source Models*           |                 |
> | LLaVA-1.5-7B                   | 48.7            |
> | LLaVA-1.6-13B                  | 61.8            |
> | SEAL                           | 74.8            |
> | Qwen2.5-7B-VL                  | 78.0 (±3.1)     |
> | **Multi-t ViGoRL-7B (Ours)** | **86.4 (±2.9)** |
> | Qwen2.5-3B-VL                  | 74.2 (±3.0)     |
> | Single-t ViGoRL-3B (Ours)    | 79.1 (±3.0)     |
> | **Multi-t ViGoRL-3B (Ours)** | **81.2 (±2.9)** |
>
> Visual search training data: We curate 11k question-answer pairs
> over small objects from Segment Anything, using GPT-4o given an object mask and scripted filtering to generate fine-grained ⟨image, question, choices, answer⟩ tuples.
>
> ---
>
> > **Reviewer comment:** SOTAs performance is not demonstrated in this work.
>
> To the best of our knowledge, ViGoRL matches or exceeds prior SOTA results across several key benchmarks, as outlined below. If we have overlooked a relevant baseline, we would welcome including it in additional comparisons.
>
> That said, our core contribution extends beyond outperforming existing models: we demonstrate that grounded reasoning provides consistent gains over standard SFT and vanilla (ungrounded) GRPO across diverse reasoning domains, supported by extensive ablations and behavioral analyses.
>
> * **VisualWebArena:** Our 7B ViGoRL model outperforms the previous 7B SOTA (ICAL) by +3.0%, without relying on human reasoning annotations or HTML inputs, which ICAL requires.
> * **V\*Bench:** As shown in Table R4 above, ViGoRL establishes SOTA among 7B-scale reasoning models (as of May 1, 2025).
> * **Web Grounding & Spatial Reasoning:** ViGoRL approaches the performance of leading commercial models such as Kimi-VL, while being fully open-source. We release all code, models, and data in the submission supplemental.
>
> > **Reviewer comment:** OOD generalization should be further improved.
>
> ViGoRL achieves strong OOD generalization compared to SFT and vanilla GRPO baselines. That said, we will add more comparisons or specific experiments if the reviewer has suggestions.
>
> We further validated OOD generalization on benchmarks with minimal training overlap. **Table R5 below (also in Supplemental Table A7) shows ViGoRL maintains or improves performance on OOD benchmarks (MMMU, RealWorldQA, V\*Bench), addressing concerns about overfitting or negative transfer.** We discuss potential extensions for even stronger OOD performance in Section 6 and limitations.
>
> ---
>
> ### Table R5: Accuracy on Out-of-Distribution (OOD) Benchmarks
>
> | **Model**                       | **MMMU**       | **RealWorldQA** | **V\*Bench**     |
> | ------------------------------- | -------------- | --------------- | ---------------- |
> | Qwen2.5-VL-3B Base              | 47.4% (±3.26) | 55.6% (±3.52)  | 74.2% (±6.22)   |
> | **ViGoRL-3B (Ours, SAT-2)**   | 46.4% (±3.26) | 60.6% (±3.46)  | 74.8% (±6.15) |
> | **ViGoRL-3B (Ours, UGround)** | 47.5% (±3.26) | 57.5% (±3.50)  | 75.1% (±6.16) |
>
> ---
>
> > **Reviewer comment:** It is required to provide to the zero-shot performance of the teacher model.
>
> We have added the zero-shot performance of our teacher model (Qwen2.5-VL-72B) in the revised manuscript, with results provided in Table R6 below. Despite using only 1k examples for distillation before RL, ViGoRL-7B outperforms Qwen2.5-VL-72B by 6.0% on SAT-2 while being 10× smaller, and closes the gap across most benchmarks.
>
> ---
>
> ### Table R6: Comparison to Qwen2.5-72B-VL
>
> | **Model**      | **SAT-2** | **BLINK** | **RoboSpatial** | **ScreenSpot-V2** | **ScreenSpot-Pro** |
> | -------------- | --------- | --------- | --------- | ----------------- | ------------------ |
> | Qwen2.5-VL-72B | 61.5% (±1.5)  | 59.5% (±2.3)  |  55.7% (±6.5)  | 88.1 (±1.8)          | 34.2 (2.6±)           |
> | ViGoRL-7B      | 67.5% (±1.5)  | 54.1% (±2.3)  | 76.4% (±7.5)  | 91.0% (±1.6)          | 33.1% (±2.3)            |

---

> > ### Comment · Reviewer_Dj1B · 2025-08-05
> > **Thanks**
> >
> > Thanks for the detailed response and extensive experiments. Most of my concerns have been addressed and I will keep the positive score.

---

> > > ### Author Response · Authors · 2025-08-06
> > >
> > > Thank you for your positive feedback and for maintaining your positive score. We greatly appreciate your constructive comments and are glad that we could address your concerns.

---

### Official Review · Reviewer_r46c · 2025-06-30

**Clarity:** 3
**Significance:** 3
**Originality:** 2
**Rating:** 5
**Confidence:** 4

**Summary:**

This paper proposes a visually grounded reinforcement learning (ViGoRL) paradigm to enhance the visual reasoning capability of multimodal LLMs. Instead of solely generating textual reasoning thoughts, ViGoRL infers natural language CoT together with the corresponding spatial grounding location in the given image. This visual reasoning process mimics the cognitive behavior of humans and enriches the conventional CoT with spatially grounded cues. The proposed ViGoRL method improves the spatial reasoning and GUI understanding abilities of MLLMs. Experimental results on various benchmarks show the effectiveness of this proposed method.

**Questions:**

1. Why would directly combining data sources of spatial reasoning and web understanding for RL training benefit all tasks?
2. What are the contributions and effectiveness of each proposed reward item?
3. What is the motivation for jointly considering the visual and textual thoughts instead of solely adopting visual CoT for visual tasks?

**Ethical Concerns:**

["NO or VERY MINOR ethics concerns only"]

**Final Justification:**

Thanks to the authors for your detailed response. My original concerns are well addressed. I will increase the score from 4 to 5.

**Limitations:**

Yes

**Quality:**

2

**Strengths And Weaknesses:**

Strengths
1. Grounding chain-of-thought reasoning into visual scenes is crucial and practical for multimodal reasoning tasks. It could improve the ability to handle tasks that require spatial understanding and capture the image regions that are relevant to the question.
2. The proposed multi-trun reinforcement learning scheme with data generation pipeline is reasonable and aligns with the practical needs to address the challenges in traditional GRPO.
3. The overall paper is easy to follow and well organized.

Weaknesses
1. In lines 263-270, the training data is curated from different sources, including spatial reasoning, web grounding, and web action prediction. However, the domain of spatial reasoning is distinct from the data of web understanding. Why would directly combining these data sources for RL training benefit all tasks? There seems to be no evidence or analysis in the current draft.
2. In lines 245-252, the reward combines multiple terms and introduces several rewards, like the grammar reward and the diversity bonus. However, there is no detailed ablation study to analyze the effectiveness and contributions of each reward item.
3. Previous work, like Argus (CVPR’25), also aims to adopt visual CoT to enhance the capability of MLLMs for visual-centric tasks. Compared with Argus (CVPR’25), the proposed ViGoRL jointly considers textual and visual CoTs. It would be better to clarify the motivation for jointly adopting visual and textual thoughts, not only applying visual CoT for the visual/spatial QA tasks.

Reference:
Argus (CVPR'25): ARGUS: Vision-Centric Reasoning with Grounded Chain-of-Thought. CVPR 2025

---

> ### Author Rebuttal · Authors · 2025-07-31
>
> Thank you for your thoughtful review and for recognizing our work's novel approach to grounding chain-of-thought reasoning in visual scenes, finding it "crucial and practical for multimodal reasoning tasks." We appreciate your positive assessment of our multi-turn reinforcement learning scheme and data generation pipeline as reasonable and well-aligned with practical needs.
>
> We have addressed your concerns through additional experiments and analysis: (1) We conducted ablations showing that removing textual reasoning causes a performance drop, demonstrating that visual and textual CoTs serve complementary roles, (2) We clarified that our training considers seperated datasets, with experiments showing minimal impact from dataset mixing, and (3) We performed comprehensive reward ablations revealing that each component contributes meaningfully to overall performance.
>
> **We present our answers in detail below and hope we’ve addressed your concerns. Please let us know if anything remains unclear. We would be happy to provide further clarification.**
>
> > **Reviewer comment:** Previous work, like Argus (CVPR’25), also aims to adopt visual CoT to enhance the capability of MLLMs for visual-centric tasks. What is the motivation for jointly considering the visual and textual thoughts instead of solely adopting visual CoT for visual tasks?
>
> Thank you for the thoughtful question. To directly test the role of textual reasoning, we ran an ablation in which we removed all textual thoughts during both the SFT and RL stages, effectively reducing it to a visual CoT-style agent. **As shown in Figure R2 below, this visual-only CoT resulted in a performance drop of 1.2% on V∗Bench, indicating that grounding alone, while helpful, does not fully substitute for the explicit composition and referencing afforded by textual reasoning.**
>
> Our motivation for jointly using both visual and textual chains of thought stems from the observation that they serve complementary roles. Visual grounding and cropping provide perceptual feedback -- what the model “sees” --  while **textual reasoning supports binding and compositional inference over these grounded regions.** Much as in human cognition, this separation allows the model not only to explore regions but also to integrate and manipulate visual evidence in a structured way, yielding more interpretable and accurate reasoning trajectories.
>
> That said, we agree with the reviewer’s broader point: we do not claim that explicit natural language reasoning is the only or necessary mechanism for visual region binding and manipulation. In the long term, latent reasoning may serve similar purposes with less reliance on textual supervision (e.g., \[1]). Nonetheless, our approach offers a practical scaffold for structured supervision, demonstrates the value of grounded reasoning, and delivers immediate performance gains. We will clarify this distinction in the revised discussion and better articulate why language remains a practical and interpretable scaffold in our current formulation.
>
> \[1] Hao et al. (2024), Training Large Language Models to Reason in a Continuous Latent Space
>
> ---
>
> ### Table R2: Multi-turn ablations
>
> | **Condition**                  | **V\*Bench**    |
> | ------------------------------ | --------------- |
> | Qwen2.5-3B-VL                  | 74.2 (±3.0)     |
> | Single-t ViGoRL-3B (Ours)    | 79.1 (±3.0)     |
> | **Multi-t ViGoRL-3B (Ours)** | **81.2 (±2.8)** |
> | └─ w/o diversity reward        | 78.0 (±3.0)     |
> | └─ w/o text thoughts           | 80.0 (±2.9)     |
>
> ---
>
> > **Reviewer comment:** Why would directly combining data sources of spatial reasoning and web understanding for RL training benefit all tasks?
>
> Thank you for this question about our training strategy. To clarify, we do not combine datasets during training and each model is trained on task-specific data. We will revise lines 263-270 to clarify this point.
>
> To quantify the effect of combining data, we conducted an experiment combining spatial and web datasets during both warm-start and RL phases. The results, shown in Table R3, reveal:
>
> - Web grounding: Small decrease (-0.2% on ScreenSpot-Pro, -0.3% on ScreenSpot-V2)
> - Spatial reasoning: Improvement of +1.1% on SAT-2
> - Both approaches significantly outperform vanilla GRPO (+12-15% SAT-2)
>
> These findings indicate that combining data has small effects on relative performance gains from our method, with slight benefits for spatial reasoning at the cost of web grounding performance.
>
> ---
>
> ### Table R3: Training with combined vs. separate datasets
>
> | **Condition**             | **ScreenSpot-V2/Pro** | **SAT-2** |
> | ------------------------- | --------- | --------------------- |
> | Qwen2.5-3B-VL             |  9.3% (±1.4) / 68.4% (±2.6)       |   46.1% (±1.5)                    |
> | Vanilla GRPO (seperate spatial or web)             |  29.0% (±2.2)  / 84.4% (±2.0)        |    50.0% (±1.6)                    |
> | ViGoRL-3B (seperate spatial or web)     | 31.1% (±2.3) / 86.5% (±1.9)   |  62.9% (±1.5)                     |
> | ViGoRL-3B (combined spatial+web) | 30.9% (±2.3) / 86.2% (±2.0)  |    64.0 (±1.5)                   |
>
> ---
>
> > **Reviewer comment:** What are the contributions and effectiveness of each proposed reward item?
>
> We thank the reviewer for this valuable suggestion. We have ablated all components of our reward formulations in Tables R4 and R5 below.
>
> Specifically:
> - Removing the multi-turn diversity bonus reward leads to a decrease of 3.2% on V\*bench. Appendix Figure A1 shows that removing this term leads to turn collapse.
> - Removing the format reward leads to a decrease of 0.6% on SAT-2.
> - Removing the accuracy reward leads to a decrease of 2.3% on SAT-2.
>
> We will include these ablations in Section 5.2 of the revised manuscript.
>
> ---
>
> ### Table R4: Multi-turn reward ablation
>
> | **Condition**                  | **V\*Bench**    |
> | ------------------------------ | --------------- |
> | Qwen2.5-3B-VL                  | 74.2 (±3.0)     |
> | Single-t ViGoRL-3B (Ours)    | 79.1 (±3.0)     |
> | **Multi-t ViGoRL-3B (Ours)** | **81.2 (±2.8)** |
> | └─ w/o diversity reward        | 78.0 (±3.0)     |
>
> ---
>
> ### Table R5: Additional reward ablations
>
> | **Condition**          | **SAT-2**    |
> | ---------------------- | ------------ |
> | ViGoRL-3B (Ours)     | 62.9% (±1.5) |
> | └─ w/o format reward   |    62.3% ± (±1.5)          |
> | └─ w/o accuracy reward |      60.6% ± (±1.5)        |

---

> > ### Comment · Reviewer_r46c · 2025-08-06
> >
> > Thanks to the authors for your detailed response. My original concerns are well addressed. I will increase the score from 4 to 5.

---

> > > ### Author Response · Authors · 2025-08-06
> > >
> > > Thank you for taking the time to review our response and for increasing your score. We are glad we could address your concerns. Your constructive feedback has been valuable in improving our work.

---

### Official Review · Reviewer_jnnD · 2025-07-02

**Clarity:** 4
**Significance:** 3
**Originality:** 3
**Rating:** 5
**Confidence:** 3

**Summary:**

This paper introduces a method for training vision-language models (VLMs) to ground their reasoning steps in visual information. Previous approaches that directly apply reinforcement learning to base models yield biased and ungrounded models regarding visual information. The proposed approach first performs supervised fine-tuning (SFT) equipped with Monte Carlo Tree Search (MCTS), then applies reinforcement learning to scale. This enables VLMs to focus not only on abstract reasoning steps but also on visually-grounded thoughts. Experiments demonstrate performance gains achieved by the introduced grounding method.

**Questions:**

- Diversity bonus (+0.2): How was this value determined?
- Computational Cost: How expensive is the introduced SFT component?

Minors:
- Table 2 is referenced before Table 1, which creates an awkward reading flow.

**Ethical Concerns:**

["NO or VERY MINOR ethics concerns only"]

**Final Justification:**

The paper is well-motivated and clearly written. The technical aspect is solid, and experiments are extensively conducted. Thus, I will keep my positive rating.

**Limitations:**

While limitations are discussed in the appendix, they should also be addressed in the main text for better visibility.

**Paper Formatting Concerns:**

Not found.

**Quality:**

4

**Strengths And Weaknesses:**

### Strengths
The paper is well-motivated and clearly written, making it easy to follow and understand the core ideas. Each component of the technical contribution is introduced effectively by explaining its necessity and addressing specific problems, which significantly enhances readability. Overall, the work is technically solid and well-written. The community will benefit from the insights and results presented in this paper.

### Weaknesses
- Scope Clarification: The problem scope of the method is not clearly defined. While the introduced SFT/RL method enhances performance on target tasks, it may degrade performance on other tasks. A clarification of the method's scope would benefit future readers.
- Limited Model Evaluation: Experiments consistently use Qwen-2.5-VL as the base model. The authors should justify this design choice and explain why other model options are not evaluated.
- Failure Analysis: The paper demonstrates well how the proposed method works when successful, but lacks discussion of failure cases and scaling bottlenecks. Understanding when and why the method fails would help readers better grasp the paper's scope and limitations.

---

> ### Author Rebuttal · Authors · 2025-07-31
>
> Thank you for your thoughtful review and for recognizing our work as well-motivated, clearly written, and technically solid. We appreciate your positive assessment of our grounded reasoning approach and its benefits to the community.
>
> We have carefully tried to address your concerns through additional experiments and analysis: (1) We clarify ViGoRL's scope, (2) We justify our choice of Qwen-2.5-VL, (3) We add comprehensive error analysis, (4) We provide ablation results on our diversity bonus, and (5) We detail computational costs of our SFT component.
>
> **We present our answers in detail below and hope we’ve addressed your concerns. Please let us know if anything remains unclear. We would be happy to provide further clarification.**
>
> > **Reviewer comment:** The problem scope of the method is not clearly defined. Training may degrade performance on other tasks.
>
> We will clarify ViGoRL's scope in Section 3.1:
> “ViGoRL targets improved performance on specific visual reasoning domains (e.g., spatial understanding, web element grounding) by fine-tuning domain-relevant reasoning skills, while preserving performance on related tasks by leveraging transferable grounding and reasoning mechanisms.”
>
> To further clarify and verify the scope:
>
> - Table R1 (also in Supplemental Table A7) shows ViGoRL maintains or improves performance on out-of-domain benchmarks (MMMU, RealWorldQA, V\*Bench), suggesting that its grounded reasoning strategies are transferable and do not result in overfitting or negative transfer.
>
> - In Table R2, we also evaluate mixed-task training (spatial + web) and find only minor trade-offs, indicating that ViGoRL’s improvements are not narrowly scoped to a single domain.
>
> - As suggested by reviewer vmK2, we train and evaluate on the open-ended domain of V\*Bench, finding ViGoRL significantly outperforms strong baselines in this additional domain (Table R3).
>
> ---
>
> ### Table R1: Accuracy on Out-of-Distribution (OOD) Benchmarks
>
> | **Model**                       | **MMMU**       | **RealWorldQA** | **V\*Bench**     |
> | ------------------------------- | -------------- | --------------- | ---------------- |
> | Qwen2.5-VL-3B Base              | 47.4% (±3.2) | 55.6% (±3.5)  | 74.2% (±6.2)   |
> | **ViGoRL-3B (Ours, SAT-2)**   | 46.4% (±3.2) | 60.6% (±3.4)  | 74.8% (±6.1) |
> | **ViGoRL-3B (Ours, UGround)** | 47.5% (±3.2) | 57.5% (±3.5)  | 75.1% (±6.1) |
>
> ---
>
> ### Table R2: Training with combined vs. separate datasets
>
> | **Condition**             | **ScreenSpot-V2/Pro** | **SAT-2** |
> | ------------------------- | --------- | --------------------- |
> | Qwen2.5-3B-VL             |  9.3% (±1.4) / 68.4% (±2.6)       |   46.1% (±1.5)                    |
> | Vanilla GRPO (seperate spatial or web)             |  29.0% (±2.2)  / 84.4% (±2.0)        |    50.0% (±1.6)                    |
> | ViGoRL-3B (seperate spatial or web)     | 31.1% (±2.3) / 86.5% (±1.9)   |  62.9% (±1.5)                     |
> | ViGoRL-3B (combined spatial+web) | 30.9% (±2.3) / 86.2% (±2.0)  |    64.0 (±1.5)                   |
>
> ---
>
> ### Table R3: Accuracy for visual search on V\*Bench
>
> | **Model Name**                 | **V\*Bench**    |
> | ------------------------------ | --------------- |
> | *Proprietary Models*           |                 |
> | Gemini-Pro                     | 48.2            |
> | GPT-4V                         | 55.0            |
> | GPT-4o                         | 66.0            |
> | *VLM Tool-Using Pipelines*     |                 |
> | VisProg                        | 41.4            |
> | VisualChatGPT                  | 37.6            |
> | MM-React                       | 41.4            |
> | Sketchpad-GPT-4o               | 80.3            |
> | IVM-Enhanced GPT-4V            | 81.2            |
> | *Open-source Models*           |                 |
> | LLaVA-1.5-7B                   | 48.7            |
> | LLaVA-1.6-13B                  | 61.8            |
> | SEAL                           | 74.8            |
> | Qwen2.5-7B-VL                  | 78.0 (±3.1)     |
> | **Multi-t ViGoRL-7B (Ours)** | **86.4 (±2.9)** |
> | Qwen2.5-3B-VL                  | 74.2 (±3.0)     |
> | Single-t ViGoRL-3B (Ours)    | 79.1 (±3.0)     |
> | **Multi-t ViGoRL-3B (Ours)** | **81.2 (±2.9)** |
>
> Visual search training data details: We curate 11k question-answer pairs
> over small objects from Segment Anything, using GPT-4o given an object mask and scripted filtering to generate fine-grained ⟨image, question, choices, answer⟩ tuples.
>
> ---
>
> > **Reviewer comment:** The authors should justify using Qwen-2.5-VL as a design choice.
>
> Thank you for raising this important point about model selection. We chose Qwen2.5-VL as our primary base model based on both its technical and practical suitability for our setup. Qwen2.5-VL was one of the strongest open-source VLMs available at the time of our experiments, with strong visual grounding capabilities already incorporated during pretraining. It also comes in multiple sizes (3B, 7B, 72B), which enabled controlled scaling and distillation within a consistent architecture and tokenizer.
>
> To address your concern about evaluating other models, we are currently running our pipeline with Kimi-VL using the EasyR1 framework, and these results will be included in the final release.
>
> ---
>
> > **Reviewer comment:** The paper lacks discussion of failure cases.
>
> Thank you for the suggestion. We now include Table R4, which categorizes SAT-2 errors. 60% arise from perceptual or task-level challenges, while 40% reflect residual reasoning flaws (e.g., overthinking, truncation, hallucination). We believe scaling or denser rewards could address aspects of both, with further discussion in Section 6 and limitations. We will include this analysis and additional qualitative failure cases in the revised manuscript.
>
> ---
>
> ### Table R4: ViGoRL Error Analysis by Type on SAT-2
>
> | **Error Type**        | **Percentage** | **Description**                                                                                                                                   |
> |-----------------------|----------------|---------------------------------------------------------------------------------------------------------------------------------------------------|
> | Excessive Reasoning   | 20%            | The model overthinks the problem.                   |
> | Incomplete Reasoning  | 10%            | The model truncates its reasoning process.                                            |
> | Hallucination         | 10%            | The model invents visual elements.                                          |
> | Challenging Visual QA | 30%            | The question or image presents inherently subtle cues, causing reasoning errors.     |
> | Base Perception       | 30%            | The underlying model makes perceptual or conceptual grounding errors.|
>
> #### Example R1: Overthinking
>
> > Question: “Is there a pot in the image?”
>
> > Image: \<Image of stove with a large pot and pan on it>
>
> *Model response:*
> \<think>
> To determine if there's a pot, I'll check the regions where pots typically appear. At (300, 600), there's a pot-like object with a handle and a lid, suggesting it could be a pot. To confirm the presence of a pot, I'll check another region. At (700, 1000), there's no pot-like object present, indicating that the previous observation was accurate. Now, I'll verify if there are any other potential pots in the image. At (500, 800), there's no pot-like object present, which supports the previous findings. Next, let's examine the area at (400, 900) to ensure there's no hidden pot. There's still no pot-like object visible here, reinforcing the previous observations. I'll now check the lower region at (600, 1200) to confirm there are no additional pots. At (600, 1200), no pot-like object is visible here, further supporting the absence of a pot in the image. Lastly, I'll check the upper region at (300, 200) to ensure there's no hidden pot. At (300, 200), no pot-like object is visible here, confirming the pot is at (300, 600) in the image.
> \</think>
>
> ---
>
> > **Reviewer comment:** How was diversity bonus determined?
>
> The diversity bonus is treated as a hyperparameter. We ablate the diversity bonus in Table R5. Removing the multi-turn diversity bonus reward leads to a decrease of 3.2% on V\*bench. We found 0.2 to encourage turn exploration while not overpowering accuracy reward.
>
> ---
>
> ### Table R5: Multi-turn reward ablation
>
> | **Condition**                  | **V\*Bench**    |
> | ------------------------------ | --------------- |
> | **Multi-t ViGoRL-3B (Ours)** | **81.2 (±2.8)** |
> | └─ w/o diversity reward        | 78.0 (±3.0)     |
>
> ---
>
> > **Reviewer comment:** How expensive is the introduced SFT component?
>
> The introduced SFT stage, including MCTS-based data generation and supervised finetuning, incurred approximately 224 GPU-hours (192 for MCTS, 32 for SFT), while the RL stage required 576 GPU-hours. Thus, SFT accounts for ~39% of the RL computational cost. It is intentially lightweight (1.5k prompts) and provides structured, grounded reasoning behaviors that RL alone fails to induce (see Table 4 and Section 5.3).
>
> Incurring additional resources for SFT before RL is in-fact common among post-training recipes [1,2]. However, unlike prior warm-start approaches that rely on large-scale human prompt curatation or annotations [1,2], our warm start uses model+search to generate reasoning traces without human supervision.
>
> [1] DeepSeek-AI. (2025), DeepSeek-R1
>
> [2] Kimi Team (2025), Kimi-VL Technical Report
>
> > Table 2 is referenced before Table 1.
>
> Thank you for the suggestions. We will revise the text to reference tables in the correct order for readability.
>
> > Limitations should be addressed in the main text.
>
> We will move the key limitations from the appendix to the main text to ensure they are clearly visible to readers.

---

> > ### Comment · Reviewer_jnnD · 2025-08-05
> >
> > I thank the authors for the detailed rebuttal. It clarified most of my major concerns and thus will keep my positive score.

---

> > > ### Author Response · Authors · 2025-08-06
> > >
> > > Thank you for maintaining your positive assessment of our work. We're glad the rebuttal successfully addressed your concerns and appreciate your thoughtful engagement with our paper.

---

### Official Review · Reviewer_vmK2 · 2025-07-03

**Clarity:** 4
**Significance:** 4
**Originality:** 3
**Rating:** 5
**Confidence:** 5

**Summary:**

This paper introduces ViGoRL (Visually Grounded Reinforcement Learning), a method that trains vision-language models to perform visual reasoning by anchoring each reasoning step to specific spatial coordinates in images. The authors argue that current VLMs rely primarily on abstract, ungrounded reasoning rather than systematic visual region analysis. Their approach combines Monte Carlo Tree Search (MCTS) to generate high-quality grounded reasoning trajectories for supervised fine-tuning, followed by Group Relative Policy Optimization (GRPO) to reinforce correct grounded sequences. The method is evaluated on spatial reasoning benchmarks (SAT-2, BLINK, RoboSpatial) and web grounding tasks (ScreenSpot, VisualWebArena), showing consistent improvements over baselines. The paper also includes behavioral analysis demonstrating that grounding amplifies visual cognitive behaviors like region exploration, subgoal setting, and backtracking.

**Questions:**

1. How does ViGoRL perform on more open-ended visual reasoning tasks that don't have clear spatial structure (e.g., abstract visual reasoning, artistic interpretation)? Are there task categories where explicit grounding might be detrimental? For example, how's the performance on V* benchmark [a]?

2. Why fixed crop size? Recent methods like CoF[b] and DeepEyes[c] adopt adaptive sizes for visual thinking. How does the proposed method compare to these methods? Are there any special concerns that hinder the authors from adaptive resolution?

3. You mention reward hacking as a limitation. Have you experimented with intermediate rewards for reasoning step quality or grounding accuracy? How might this affect the training dynamics?

[a] V*: Guided Visual Search as a Core Mechanism in Multimodal LLMs

[b] Chain-of-Focus: Adaptive Visual Search and Zooming for Multimodal Reasoning via RL

[c] DeepEyes: Incentivizing "Thinking with Images" via Reinforcement Learning

**Ethical Concerns:**

["NO or VERY MINOR ethics concerns only"]

**Final Justification:**

My concerns are addressed, and I choose to keep my positive rating.

**Limitations:**

Yes, the authors have adequately addressed limitations in Appendix A2, including reward hacking issues, computational considerations, and the need for intermediate rewards. They also provide thoughtful discussion of broader impacts in A3, covering both positive applications (medical imaging, accessibility) and potential negative uses (surveillance). The acknowledgment of dataset bias and the need for human oversight in sensitive applications demonstrates responsible consideration of societal impact.

**Quality:**

4

**Strengths And Weaknesses:**

## Strengths

1. Novel and Well-Motivated Approach: The core insight that explicit spatial grounding can improve visual reasoning is compelling and well-supported by cognitive science literature. The connection to human visual attention patterns provides strong theoretical motivation.

2. Rigorous Analysis of Visual Thinking: The paper provides an exceptionally thorough behavioral analysis of visual reasoning in VLMs, systematically quantifying cognitive behaviors such as region exploration, visual subgoal setting, verification, and backtracking. This rigorous examination of how models actually "think" visually offers valuable insights to the community about the internal mechanisms of visual reasoning and represents a significant methodological contribution beyond the core technical approach.

3. Comprehensive Experimental Design: The paper includes thorough evaluations across multiple benchmarks, proper ablation studies, behavioral analysis, and human evaluation. The behavioral coding methodology following Gandhi et al. is particularly valuable for understanding what the model learns.

4. Strong Empirical Results: Consistent improvements across all benchmarks are impressive, with particularly notable gains on SAT-2 (+12.9 points over vanilla GRPO) and competitive performance on web grounding tasks.

## Weaknesses:

1. Reward Design Limitations: The reliance on final-answer rewards may lead to reward hacking, as acknowledged by the authors. The format reward for valid coordinates is somewhat simplistic and may not capture the quality of grounding.

2. Limited Analysis of Failure Cases: While the paper shows when grounding helps, there's insufficient analysis of when it might hurt or be unnecessary. Some tasks may not benefit from explicit spatial grounding.

3. Generalization Questions: Most evaluations are on relatively structured tasks (spatial reasoning, GUI interaction). It's unclear how well the approach generalizes to more open-ended visual reasoning tasks.

---

> ### Author Rebuttal · Authors · 2025-07-31
>
> Thank you for your thoughtful feedback. We were encouraged you found our work novel with compelling theoretical grounding in cognitive science, appreciated our "exceptionally thorough behavioral analysis" as a "significant methodological contribution", and recognized our comprehensive experimental design and "impressive" empirical results with consistent improvements across all benchmarks.
>
> We have carefully addressed your concerns through additional experiments and analysis. To summarize: (1) we tested ViGoRL on V\*Bench as suggested, finding it significantly outperforms strong baselines, (2) compared adaptive vs. fixed cropping and found no performance difference, with further discussion on each approach, (3) we provide further justification and discussion on intermediate reward and mitigation of reward hacking, and (4) we clarify when grounded reasoning may be unnecessary.
>
> **We present our answers in detail below and hope we’ve addressed your concerns. Please let us know if anything remains unclear. We would be happy to provide further clarification.**
>
> > **Reviewer comment:** Most evaluations are on relatively structured tasks (spatial reasoning, GUI interaction). How's the performance on V* benchmark?
>
> Thank you for raising this point about testing the method on more open-ended visual reasoning tasks. **As suggested, we have now run ViGoRL on V\*Bench**, which tests open-ended visual search including open-ended question answering about fine-grained image details.
>
> We show the results in Table R1. Our key findings are:
>
> - ViGoRL-7B achieves 86.4%, **outperforming open-source reasoning models (as of May 1, 2025) and surpassing GPT-4o by 20.4%**.
> - Even our 3B model reaches 81.2%, matching sophisticated tool-using pipelines like IVM-Enhanced GPT-4V
> - Multi-turn ViGoRL improves performance over the single-turn variant by 2.1%, demonstrating the utility of multi-turn zoomed-in feedback in this visual search settings.
>
> This strong performance on visual search reasoning tasks demonstrates that grounded reasoning generalizes beyond spatial and web domains.
>
> ---
>
> ### Table R1: Accuracy for visual search on V\*Bench
>
> | **Model Name**                 | **V\*Bench**    |
> | ------------------------------ | --------------- |
> | *Proprietary Models*           |                 |
> | Gemini-Pro                     | 48.2            |
> | GPT-4V                         | 55.0            |
> | GPT-4o                         | 66.0            |
> | *VLM Tool-Using Pipelines*     |                 |
> | VisProg                        | 41.4            |
> | VisualChatGPT                  | 37.6            |
> | MM-React                       | 41.4            |
> | Sketchpad-GPT-4o               | 80.3            |
> | IVM-Enhanced GPT-4V            | 81.2            |
> | *Open-source Models*           |                 |
> | LLaVA-1.5-7B                   | 48.7            |
> | LLaVA-1.6-13B                  | 61.8            |
> | SEAL                           | 74.8            |
> | Qwen2.5-7B-VL                  | 78.0 (±3.1)     |
> | **Multi-t ViGoRL-7B (Ours)** | **86.4 (±2.9)** |
> | Qwen2.5-3B-VL                  | 74.2 (±3.0)     |
> | Single-t ViGoRL-3B (Ours)    | 79.1 (±3.0)     |
> | **Multi-t ViGoRL-3B (Ours)** | **81.2 (±2.9)** |
>
> Visual search RL training data: We curate 11k question-answer pairs
> over small objects from Segment Anything, using GPT-4o given an object mask and scripted filtering to generate fine-grained ⟨image, question, choices, answer⟩ tuples.
>
> ---
>
> > **Reviewer comment:** Why fixed crop size?
>
> Thank you for this insightful question regarding adaptive cropping. We conducted an additional experiment where both MCTS warm-start and RL phases in ViGoRL use predicted bounding boxes (adaptive crops) instead of fixed-size point-based crops.
>
> **As shown in Table R2, we observe no difference in V\*Bench accuracy between the adaptive bounding box and fixed-point cropping approaches.** We attribute this to two main reasons:
>
> 1. **Sufficient context from fixed crops**: The default crop size is typically large enough to capture both the target region and its surrounding context at a high resolution for most V\*Bench examples.
>
> 2. **Multi-crop coverage of larger objects**: When the region of interest is too large for a single fixed crop, ViGoRL naturally samples multiple overlapping crops in the vicinity, analogous to human visual exploration through multiple fixations. A combination of these partial crops often suffices to infer the correct answer, especially when supported by a global view.
>
> While adaptive cropping could, in principle, be more efficient, we encountered practical limitations:
>
> * Bounding box predictions from Qwen are often imprecise.
> * Inconsistent aspect ratios from adaptive crops can introduce variability during training.
>
> We agree this is a valuable design dimension and plan to further explore trade-offs between adaptive and fixed cropping in future work.
>
> ---
>
> ### Table R2: Additional multi-turn ablations
>
> | **Condition**                  | **V\*Bench**    |
> | ------------------------------ | --------------- |
> | Qwen2.5-3B-VL                  | 74.2 (±3.0)     |
> | Single-t ViGoRL-3B (Ours)    | 79.1 (±3.0)     |
> | **Multi-t ViGoRL-3B (Ours)** | **81.2 (±2.8)** |
> | └─ w/o diversity reward        | 78.0 (±3.0)     |
> | └─ **w/ bounding box outputs**           | **81.2 (±2.8)**    |
>
> ---
>
> > **Reviewer comment:** The reliance on final-answer rewards may lead to reward hacking, as acknowledged by the authors. Have you experimented with intermediate rewards for reasoning step quality or grounding accuracy? How might this affect the training dynamics?
>
> We agree that relying solely on final-answer rewards introduces a risk of reward hacking, and we explicitly acknowledge this limitation. To mitigate this, we use a MCTS-based warm start, a KL penalty, and a structured format rewards that enforce correct tag usage and require valid grounding coordinates, acting as a minimal but effective constraint to maintain warm-start grounded outputs during training.
>
> To assess the quality of intermediate reasoning, we conducted a human evaluation (Section 5.4), where **annotators judged reasoning steps to be both spatially accurate and helpful** for understanding model behavior, demonstrating the effectiveness of our minimal design.
>
> We agree that incorporating more nuanced intermediate rewards (e.g., region relevance or semantic alignment) is a promising direction, but non-trivial: defining automated signals that robustly reflect grounding “quality” remains an open research challenge. These additional signals would require additional supervision such as human data (e.g., eye gaze) or specialized judge models. We plan to explore this in future work and will clarify this in the main paper.
>
> > **Reviewer comment:** While the paper shows when grounding helps, there's insufficient analysis of when it might hurt or be unnecessary. Some tasks may not benefit from explicit spatial grounding.
>
> We appreciate the reviewer’s point. We agree that spatial grounding may not always be necessary, particularly for tasks dominated by global scene cues or simple object recognition. Prior work shows that vision-language models like GPT-4V can perform well on such tasks without explicit region-level reasoning \[1].
>
> However, a growing body of work highlights that grounding becomes helpful for tasks involving fine-grained spatial reference, ambiguity resolution, or multi-step visual reasoning. For instance:
>
> * Grounding improves spatial reasoning accuracy, especially when simple visual attention fails to resolve spatial relations \[2].
> * Referring expression grounding benefits substantially from localized object anchoring \[3].
> * Explicitly prompting for region references improves performance on spatial QA and tool-use tasks \[4].
> * Prompting grounded intermediate steps boosts both interpretability and accuracy across visual tasks \[5].
>
> We provide an example below (Example R1) where our ViGoRL model does unnecessary grounding and verification for a recognition task. While not harming performance, this behavior may lead to excessive computation. We note that this overthinking occurs with existing reasoning models \[6].
>
> We will revise Section 6 to clarify when spatial grounding is particularly helpful, and add additional qualitative results in the appendix.
>
> \[1] OpenAI, GPT-4V System Card, 2023.
>
> \[2] Liu et al., What Makes Spatial Reasoning Hard for VLMs?, 2023.
>
> \[3] Chen et al., RefCLIP: Referring with Class Name Aware Vision-Language Models, 2022.
>
> \[4] Wu et al., V\*Bench: Evaluating Tool-augmented VLMs, 2023.
>
> \[5] Yang et al., Thinking in Pictures, 2024.
>
> \[6] Chen, X. Do not think that much for 2+3=? On the overthinking of o1-like LLMs. 2025.
>
> ---
>
> ### Example R1: Overthinking
> > Question: “Is there a pot in the image?”
>
> > Image: \<Image of stove with a large pot and pan on it>
>
> *Model response:* \<think> To determine if there's a pot, I'll check the regions where pots typically appear. At (300, 600), there's a pot-like object with a handle and a lid, suggesting it could be a pot. To confirm the presence of a pot, I'll check another region. At (700, 1000), there's no pot-like object present, indicating that the previous observation was accurate. Now, I'll verify if there are any other potential pots in the image. At (500, 800), there's no pot-like object present, which supports the previous findings. Next, let's examine the area at (400, 900) to ensure there's no hidden pot. There's still no pot-like object visible here, reinforcing the previous observations. I'll now check the lower region at (600, 1200) to confirm there are no additional pots. At (600, 1200), no pot-like object is visible here, further supporting the absence of a pot in the image. Lastly, I'll check the upper region at (300, 200) to ensure there's no hidden pot. At (300, 200), no pot-like object is visible here, confirming the pot is at (300, 600) in the image. \</think>

---

> > ### Comment · Reviewer_vmK2 · 2025-08-05
> >
> > Thank you for the detailed and thoughtful rebuttal. This successfully addresses my concerns, and I will maintain my positive score.
> >
> > I'm particularly glad to see the additional experiments on V*Bench. The results are compelling and demonstrate strong generalization beyond structured tasks. However, I noticed that the standard deviations reported for both baselines and your method appear relatively large. I would be curious to hear your perspective on this—do you believe this variance is due to the dataset characteristics, model stochasticity, or other factors?

---

> > > ### Author Response · Authors · 2025-08-06
> > >
> > > Thank you for your thoughtful feedback and for maintaining your positive score. We're delighted that you appreciated the additional V\*Bench experiments and found our results compelling.
> > >
> > > Regarding your question about the larger standard deviations in our V\*Bench results: The larger error bars are primarily due to the smaller dataset size of V\*Bench compared to our other evaluation benchmarks. V\*Bench contains fewer test samples, and since we compute error bars across these samples, the smaller sample size naturally leads to larger standard deviations. Thus, the variance we observe is primarily due to dataset characteristics (dataset size) rather than model instability or other factors.
> > >
> > > We appreciate your continued engagement with our work and are happy to provide any additional clarification if needed.

---

### Note · Authors · 2025-08-11

We sincerely thank all reviewers for their constructive feedback and positive engagement throughout the review process. We are encouraged that reviewers found our work **novel** (vmK2: "a significant methodological contribution"; r46c: "crucial and practical for multimodal reasoning tasks"), **technically solid** (jnnD: "well-written and technically solid"), and **significant to the community** (Dj1B: "significant contribution to visual grounding field").

Through extensive additional experiments during the rebuttal period, we addressed the reviewers' major concerns:

**1. Generalization & Scope:** Following vmK2's suggestion, we demonstrated strong performance on V*Bench (86.4% accuracy), **significantly outperforming GPT-4o (+20.4%) and tool-use pipelines.** vmK2 found these results "compelling and demonstrate strong generalization beyond structured tasks." OOD evaluation shows maintained/improved performance on MMMU and RealWorldQA (Dj1B).

**2. Technical Clarifications:** We provided reward ablations requested by r46c and Dj1B, showing each reward component's contribution (diversity bonus: +3.2%, format reward: +0.6%, accuracy reward: +2.3%). We clarified that models are trained on task-specific data, with minimal impact from dataset mixing (r46c). Our SFT component (39% of RL cost) provides essential reasoning that RL alone fails to induce (jnnD).

**3. Design Choices:** Per reviewer requests, we justified Qwen2.5-VL selection (jnnD) and compared adaptive vs. fixed cropping (vmK2, Dj1B), finding comparable performance. We validated our binary reward design over Euclidean distance (+2.2% improvement; Dj1B). Multi-turn feedback proves especially valuable for fine-grained tasks requiring high-resolution disambiguation (Dj1B).

**4. Failure Analysis:** We added comprehensive error analysis categorizing failure modes and clarifying when grounding may be unnecessary, addressing jnnD's concerns about scope definition.

**All reviewers explicitly confirmed their concerns were addressed.** vmK2: "successfully addresses my concerns"; jnnD: "clarified most of my major concerns"; r46c: "my original concerns are well addressed" (**increasing score 4→5**); Dj1B: "most of my concerns have been addressed." We have incorporated all suggested revisions into our manuscript and **release all code, models, and training data.** We believe ViGoRL's grounded reasoning approach represents a meaningful step toward more interpretable and capable visual reasoning systems.

---

### Decision · Program_Chairs · 2025-09-17

**Decision:**

Accept (poster)

**Comment:**

The paper introduces ViGoRL, a novel reinforcement learning framework that enhances visual reasoning in vision-language models by explicitly grounding each reasoning step in spatial image coordinates. Its core claim is that such grounding improves performance on spatial and web-based reasoning tasks by mimicking human visual attention. Strengths include a rigorous behavioral analysis, strong empirical results across multiple benchmarks (e.g., SAT-2, V*Bench), and a well-motivated methodology inspired by cognitive science. Weaknesses initially included concerns about reward design, generalization to open-ended tasks, and limited failure analysis.

During the rebuttal period, reviewers raised several key points: vmK2 questioned generalization (addressed via V*Bench results showing +20.4% over GPT-4o); jnnD requested scope clarification and failure analysis (added error categorization and OOD results); r46c sought reward ablations and multi-modal justification (provided component-wise reward contributions and textual vs. visual CoT comparisons); Dj1B queried reward design and adaptive cropping (showed binary reward outperforms Euclidean distance and fixed vs. adaptive cropping parity). All reviewers acknowledged their concerns were addressed, with two increasing scores post-rebuttal.

The ratings of this submission are 5,4,5,5, suggesting a clear recommendation of acceptance. Taking all reviews into consideration, I recommend acceptance of this paper.